# A transcription network underlies the dual genomic coordination of mitochondrial biogenesis

Fan Zhang, Annie Lee, Anna V Freitas, Jake T Herb, Zong-Heng Wang, Snigdha Gupta, Zhe Chen, Hong Xu*

National Heart, Lung, and Blood Institute, National Institutes of Health, Bethesda, United States

## eLife Assessment

This study's findings substantially advance our understanding of an **important** aspect of mitochondrial metabolism. The data are **compelling** and the study is well executed. The work is relevant to all who are interested in the biogenesis of mitochondria.

**\*For correspondence:**
hong.xu@nih.gov

**Competing interest:** The authors declare that no competing interests exist.

**Abstract** Mitochondrial biogenesis requires the expression of genes encoded by both the nuclear and mitochondrial genomes. However, aside from a handful transcription factors regulating specific subsets of mitochondrial genes, the overall architecture of the transcriptional control of mitochondrial biogenesis remains to be elucidated. The mechanisms coordinating these two genomes are largely unknown. We performed a targeted RNAi screen in developing eyes with reduced mitochondrial DNA content, anticipating a synergistic disruption of tissue development due to impaired mitochondrial biogenesis and mitochondrial DNA (mtDNA) deficiency. Among 638 transcription factors annotated in the *Drosophila* genome, 77 were identified as potential regulators of mitochondrial biogenesis. Utilizing published ChIP-seq data of positive hits, we constructed a regulatory network revealing the logic of the transcription regulation of mitochondrial biogenesis. Multiple transcription factors in core layers had extensive connections, collectively governing the expression of nearly all mitochondrial genes, whereas factors sitting on the top layer may respond to cellular cues to modulate mitochondrial biogenesis through the underlying network. CG1603, a core component of the network, was found to be indispensable for the expression of most nuclear mitochondrial genes, including those required for mtDNA maintenance and gene expression, thus coordinating nuclear genome and mtDNA activities in mitochondrial biogenesis. Additional genetic analyses validated YL-1, a transcription factor upstream of CG1603 in the network, as a regulator controlling CG1603 expression and mitochondrial biogenesis.

## Introduction

Mitochondria respiration, carried out by the electron transport chain (ETC) complexes, converts the energy in chemical fuels to the electrochemical potential across the mitochondrial inner membrane ($\Delta\psi_m$) that drives the synthesis of ATP. Deficient ETC not only impairs energy metabolism, but may also disrupt cellular redox balance and various biosynthetic pathways (*Shen et al., 2022*; *Spinelli and Haigis, 2018*), and is associated with various human diseases (*Gorman et al., 2016*). Mitochondria are under dual genetic control. Their own genome, mitochondrial DNA (mtDNA) encodes 13 core subunits of ETC, alongside 2 rRNAs and 22 tRNAs that are required for the translation of these protein-coding genes inside the mitochondrial matrix (*Chen et al., 2019*).

The majority of more than 1000 mitochondrial proteins including the remaining ETC subunits, factors for mtDNA replication and transcription, and mitochondrial ribosomal proteins are all encoded on the nuclear genome (*Hock and Kralli, 2009*; *Taylor and Turnbull, 2005*). The mitochondrial transcription factor A (TFAM) compacts mtDNA into nucleoids and is a key regulator of mtDNA copy number (*Alam et al., 2003*; *Scarpulla, 2008*). TFAM, together with other auxiliary factors including mtTFB1 and mtTFB2, promotes the transcription of mtDNA by mitochondrial RNA polymerase (POLRMT) into long polycistronic precursor RNAs, which are further processed into individual RNAs (*Chen et al., 2019*; *Falkenberg et al., 2002*). POLRMT can also generate short RNA oligos, owing to its exoribonuclease activity, to prime mtDNA replication by polymerase γ (*Liu et al., 2022*). Nuclear-encoded mitochondrial proteins are synthesized in the cytoplasm and imported into mitochondria (*Wiedemann and Pfanner, 2017*). Hence, mitochondrial biogenesis is influenced by the abundance and activities of mitochondrial translocases as well. The intricate interplay between mitochondrial and nuclear-encoded components demands coordinated activities of these two genomes, to maintain the efficiency and the integrity of oxidative phosphorylation system and other critical mitochondrial processes (*Hock and Kralli, 2009*; *Scarpulla, 2008*).

Mitochondrial respiration, particularly, the contents and the activity of ETC, is fine-tuned to cope with the developmental and tissue-specific metabolic demands (*Fernández-Vizarra et al., 2011*). Various transcriptional cascades have emerged as effective and adaptable mechanisms regulating ETC biogenesis. The nuclear respiration factors, NRF1 and NRF2, activate the expression of many nuclear-encoded ETC subunits and genes essential for mtDNA replication and transcription (*Scarpulla, 2008*). This regulation allows NRFs to indirectly control the expression of mtDNA-encoded genes, and hence coordinate the activities of both genomes in ETC biogenesis. The peroxisome proliferator-activated receptors (PPARs), upon activation by diverse lipid ligands, induce the expression of nuclear genes in fatty acids oxidation pathway (*Berger and Moller, 2002*). Another family of nuclear receptors, the estrogen-related receptors (ERRs) regulate nuclear genes involved in oxidative phosphorylation, including ETCs and the citric acid (*Deblois and Giguère, 2011*). Notably, all forementioned transcription factors (TFs) share a common co-activator, PPARγ co-activator-1α (PGC-1α), that directly stimulates the basal transcriptional machinery (*Finck and Kelly, 2007*; *Hock and Kralli, 2009*; *Scarpulla et al., 2012*). PGC-1α and related co-activators are dynamically regulated in responses to various physiological or environmental cues, to adjust metabolic program and energy metabolism accordingly (*Hock and Kralli, 2009*; *Scarpulla et al., 2012*). Members of PPARs or ERRs families often show tissue-specific expression and regulate subsets of mitochondrial genes (*Deblois and Giguère, 2011*; *Hock and Kralli, 2009*). However, given the large number and diverse evolution origins of mitochondrial genes (*Kurland and Andersson, 2000*; *Rath et al., 2021*), the current understanding of transcriptional regulation of mitochondrial biogenesis appears incomplete. Additionally, mechanisms coordinating nuclear genome and mtDNA activities in ETC biogenesis remain unclear.

Recently, the modERN (model organism Encyclopedia of Regulatory Networks) project generated genome-wide binding profiles of a large set of TFs in *Caenorhabditis elegans* and *Drosophila melanogaster* (*Kudron et al., 2018*). The global mapping of TF-DNA interactions could potentially be applied to identify the transcriptional network governing mitochondrial biogenesis. However, the DNA binding profiles of TFs have their limitations in understanding the true biological functions of TFs in gene regulation (*Jiang and Mortazavi, 2018*). Gene expression is, in most cases, subject to the combined influences of multiple TFs. Additionally, an individual TF may have either activating or repressive roles based on the local chromatin environment (*Jiang and Mortazavi, 2018*). Furthermore, despite the substantial progress in bioinformatics analyses, the interpretation of genome-wide omics data still has its limitations due to a lack of robust statistical algorithms, variations in biological contexts, and intrinsic experimental variations (*Angelini and Costa, 2014*; *Zhou et al., 2016*). The integration of the DNA binding profiles with functional genetic and genomic studies is ideal to study gene expression regulations (*Jiang and Mortazavi, 2018*; *Park, 2009*).

The *Drosophila* eye is an excellent model for genetic analyses due to its ease of assessment and minimal impact on other physiological processes in the presence of developmental abnormalities. The cell proliferation and differentiation during eye development require robust mitochondrial respiration, and adult eyes are severely disrupted by mutations affecting nuclear ETC subunits or the mitochondrial translation apparatus (*Liao et al., 2006*; *Owusu-Ansah et al., 2008*). We previously developed a genetic scheme to generate mtDNA deficiency by expressing a mitochondrially targeted restriction

enzyme (*Chen et al., 2015*; *Xu et al., 2008*). In this study, we performed an RNAi screen, targeting 638 TFs annotated in the *Drosophila* genome, in the presence of mtDNA deficiency in developing eyes. We recovered 77 TFs, RNAi against which had synergistic effects with the mtDNA deficiency in causing the small-eye phenotype. We further followed up on CG1603, one of the strongest hits from the initial modifier screen and revealed that it was essential for coordinating the nuclear genome and mtDNA for ETC biogenesis. Additional network analyses on the recovered hits using published DNA binding profiles illustrated potential regulatory connections and a complex hierarchy of the transcription regulations on mitochondrial biogenesis. The combination of genetic and bioinformatic analyses also facilitated the identification of YL-1 as an upstream regulator of CG1603.

## Results

### The design of a genetic modifier screen for genes regulating mtDNA maintenance and expression

Mutations on nuclear-encoded ETC genes block cell cycle and disrupt the differentiation and morphogenesis of developing eyes (*Mandal et al., 2005*; *Owusu-Ansah et al., 2008*). RNAi against *tfam* or *myc*, which promote the expression of ETC genes encoded on mtDNA and the nuclear genome, respectively, also reduced adult eye size (*Figure 1A*). These observations prompted us to use fly eye as a model to identify TFs regulating ETC biogenesis. However, inhibitions of any genes essential for cell viability, proliferation, or differentiation, exemplified by RNAi targeting a mitotic cyclin, *CycB* (*Figure 1A*), would disrupt eye development. Therefore, assaying eye morphology alone is not sufficient to enrich candidates regulating ETC biogenesis.

The mitochondrial genome of wild type (wt) *D. melanogaster* contains a single XhoI site. The expression of a mitochondrially targeted restriction enzyme, XhoI (MitoXhoI) in *Drosophila* ovary effectively selects for escaper progeny carrying mtDNA mutations that abolish the XhoI site (*Xu et al., 2008*). In a heteroplasmic background containing both wt and XhoI-resistant genome (*XhoI⁻*), the expression of MitoXhoI can effectively remove the wt genome and hence generate mtDNA deficiency (*Chen et al., 2015*). As a result, the adult eyes were slightly smaller than the control (*Figure 1A*). Considering that mtDNA encodes core components of ETC, we reasoned that inhibiting a gene related to ETC biogenesis would have a synergistic effect with the mtDNA deficiency on eye development, and the combination of these two genetic manipulations should lead to a stronger disruption of eye development than either of these conditions individually (*Figure 1A and B*). On this basis, we devised a scheme of modifier screen in eye for genes involved in ETC biogenesis (*Figure 1B*).

### The RNAi modifier screen identifying TFs regulating ETC biogenesis

To assess the efficacy of this scheme, we carried out a pilot RNAi screen, covering 124 nuclear-encoded mitochondrial genes and 58 non-mitochondrial genes annotated in various cellular processes (*Figure 1C–E* and *Supplementary file 1*). In practice, male flies carrying a *UAS-IR* transgene were crossed with *Sco/CyO, mitoXhoI; eyeless-GAL4* heteroplasmic female flies (carrying both wt and *XhoI⁻* mtDNA). This cross generated two groups of offspring, RNAi-only and RNAi together with MitoXhoI expression (RNAi+MitoXhoI) that were cultured in the same vial, thereby minimizing any potential discrepancy caused by environmental factors. Most RNAi flies survived to adult stage but had reduced eye size. A few RNAi flies were lethal at the pupae stage, due to a lack of head capsule that is derived from the eye-antenna disc.

For most genes tested in the pilot screen, eyes of RNAi+MitoXhoI flies were smaller than the corresponding RNAi-only flies. To rule out a simple additive effect between RNAi and MitoXhoI expression, we carried out additional analyses to semi-quantify a potential synergy between RNAi intervention and mtDNA deficiency caused by MitoXhoI expression. The eye size of progeny was arbitrarily scored on a scale from 0 to 5 (*Figure 1C*). The indexes of eye size reduction (Index-R) of RNAi and RNAi+MitoXhoI flies were calculated by normalizing the mean eye size scores of each genotype to the corresponding values of control RNAi or control RNAi+MitoXhoI, respectively, and were subsequently plotted against each other on a linear graph (*Figure 1D*). If an RNAi intervention had a synergistic effect with mtDNA deficiency, it would lie below the diagonal line. We also included *ewg*, the fly homolog of *NRF-1*, in the pilot screen to set the threshold for calling out positive hits. Of total 40 genes that are related to ETC biogenesis (Mito-EBR) including ETC subunits, mitochondrial protein import and membrane

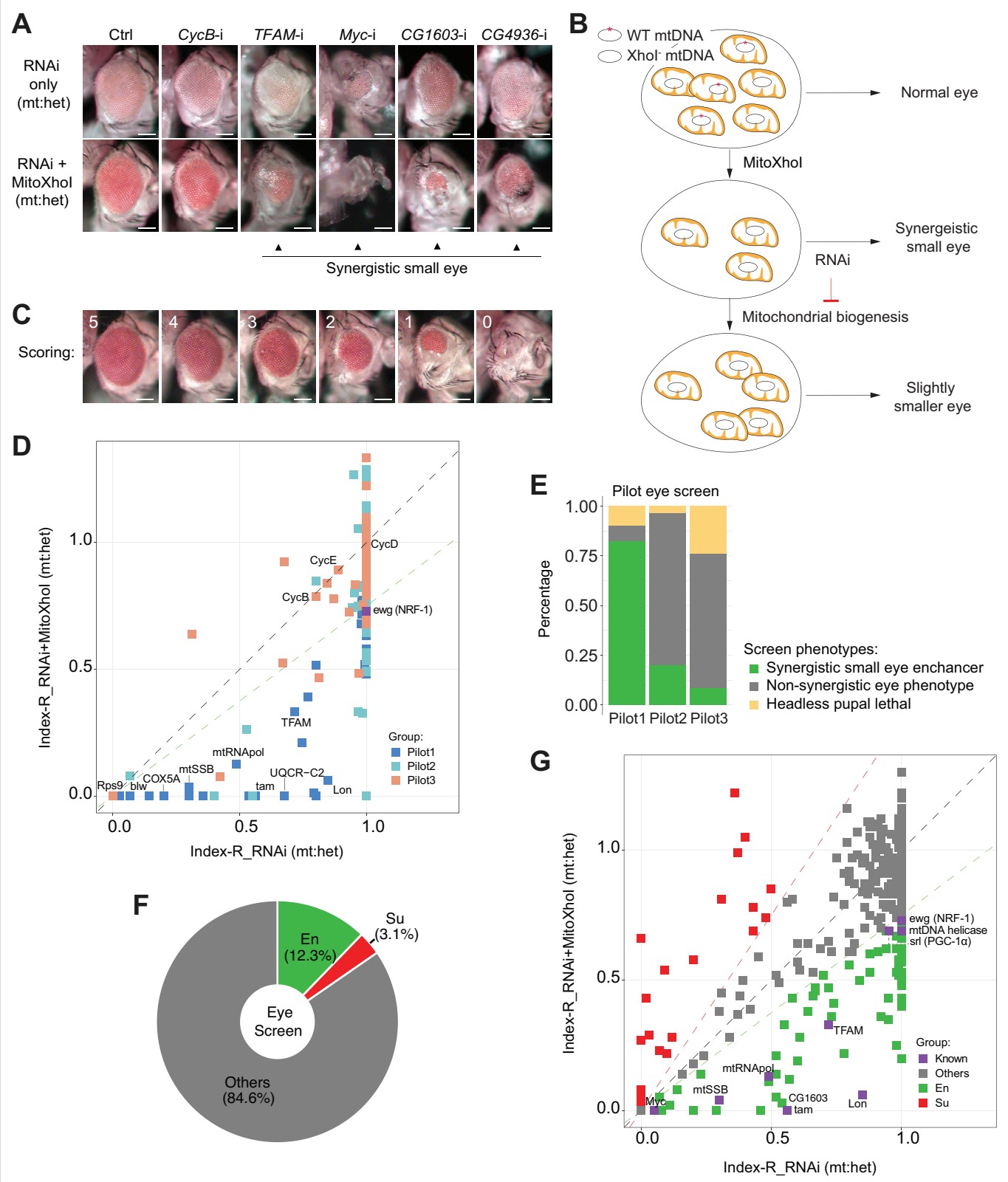

**Figure 1.** A genetic modifier screen identifying transcription factors regulating electron transport chain (ETC) biogenesis. (**A**) Representative images of adult eye of the control RNAi (Ctrl) and RNAi of selected genes tested in the eye screen, including *CycB* RNAi (*CycB*-i), *TFAM* RNAi (*TFAM*-i), *Myc* RNAi (*Myc*-i), *CG1603* RNAi (*CG1603*-i), and *CG4936* RNAi (*CG4936*-i). The upper panel shows eyes from RNAi-only offspring, and lower panel displays eyes from RNAi+MitoXhoI offspring cultured at the same condition. Arrowheads indicate the synergistic small-eye phenotype resulting from the

*Figure 1 continued on next page*

*Figure 1 continued*

combination of gene knockdown and the mitochondrial DNA (mtDNA) deficiency caused by mitoXhoI in the background of heteroplasmic mtDNAs. Scale bars: 100 μm. (**B**) Schematic of the genetic modifier screen methodology (see text for details). (**C**) Representative images illustrating the scoring of eye size. Scale bars: 100 μm. (**D**) A plot illustrating the calling of positive hits in the pilot screen. Each datapoint represents the Index-R of RNAi (X values) or RNAi+MitoXhoI flies (Y values) for each gene belonging to the different groups (see (**E**) and *Supplementary file 1* for details). Genes with datapoints below the gray diagonal dash line exhibited a synergistic effect when combining their RNAi with mtDNA deficiency suggesting a potential role in regulating ETC biogenesis. The datapoint for *ewg*, the fly homolog of *NRF-1*, is labeled in purple. The green dashed line of slope 0.75 outlines the threshold for calling out positive hits based on *ewg*'s performance in the screen. (**E**) Graph summarizing the pilot screen of nuclear-encoded genes, demonstrating the efficacy of this screen in identifying genes involved in mitochondrial ETC biogenesis. Pilot group 1 (Pilot1) has 40 genes that are either nuclear-encoded ETC subunits or related to mtDNA maintenance and gene expression (Mito-EBR). Pilot2 has 84 genes involved in other mitochondrial processes. Pilot3 has 58 essential genes from other cellular components. (**F**) Graph summarizing the percentages of synergistic enhancers (En) and suppressors (Su) identified in the screen (see (**G**) and *Supplementary file 1* for details). (**G**) A plot illustrating the calling of positive hits in the screen of transcription factor (TF) genes. Factors that are known to be involved in mitochondrial or ETC biogenesis are marked in purple (Known). The green dashed line outlines the threshold for calling out synergistic enhancers (En, green square). The red dashed line of slope 1.5 outlines the threshold for calling out suppressors (Su, red square).

The online version of this article includes the following source data for figure 1:

**Source data 1.** Raw data used to generate *Figure 1D–G*.

insertion machinery, ETC assembly factors, and proteins related to the expression of mtDNA-encoded ETC subunits, 82.5% (33 genes) emerged as enhancers (*Figure 1E*). The proportions of synergistic enhancers were much lower in the group of genes involved in other mitochondrial processes (20.2%) or the group of other essential genes not related to mitochondria (8.6%), indicating the efficacy of this modifier screen in enriching genes related to ETC biogenesis (*Figure 1E*).

To understand transcriptional regulations of ETC biogenesis, we screened 1264 RNAi lines that cover 638 genes annotated as transcriptional regulators in the *Drosophila* genome. A total of 77 enhancers were identified (*Figure 1F and G* and *Supplementary file 1*), including all known factors involved in ETC biogenesis such as Myc, TFAM (*Scarpulla, 2008*; *Wang et al., 2019*). We also recovered 20 suppressors, of which, eyes of RNAi+MitoXhoI flies were larger than the corresponding RNAi-only flies.

## Regulatory network of mitochondrial biogenesis

Among 77 TFs identified in the initial modifier screen, 49 TFs have ChIP-seq data available in modERN (*Kudron et al., 2018*). To further understand the transcriptional regulation of ETC biogenesis, we performed the network analysis on these 49 TFs using the '*vertex sort*' algorithm (*Jothi et al., 2009*), and constructed a regulatory network (*Figure 2A* and *Supplementary file 2*). All 49 TFs had binding sites on the promoter region of at least one nuclear mitochondrial gene (*Figure 2B*, *Figure 2—figure supplement 1*, and *Supplementary file 3*). Respectively, 89% nuclear-encoded mitochondrial genes (851), including nearly all Mito-EBR genes (*Supplementary file 3*), were bound by at least one TF. Given that 28 hits were not included in the analyses due to a lack of ChIP-seq data, the actual coverage of total 77 TFs on nuclear mitochondrial genes would be more comprehensive. Six TFs bound to more than half of mitochondrial genes (*Figure 2B*, *Figure 2—figure supplement 1*, and *Supplementary file 3*). However, not a single TF covered all mitochondrial genes, or all genes in a specific mitochondrial process, which is consistent with the diverse evolution origin of mitochondrial genes (*Kurland and Andersson, 2000*). It also indicates that there is no such a 'master' regulator controlling all aspects of mitochondrial genesis. Forty-seven TFs were identified as strongly connected components due to their extensive connections and were classified in the core or bottom layer of the hierarchical structure, suggesting complex co-regulations and potential redundancy among these TFs in controlling mitochondrial biogenesis (*Figure 2A* and *Supplementary file 2*). Through this interconnected network, every single node can link to all 851 mitochondrial genes in the coverage. Two TFs, Crg-1 and CG15011, were identified as the top-layer TFs with no upstream regulators in the network (*Figure 2A* and *Supplementary file 2*). Crg-1 is a circadian- regulated gene (*Rouyer et al., 1997*). CG15011 is an X-box binding protein and had a binding profile similar to another X-box binding protein, Xbp1 (*Supplementary file 3*), a stress respondent and regulator (*Acosta-Alvear et al., 2007*). These top-layer TFs may sense physiological oscillations and stresses to modulate mitochondrial biogenesis through the underlying network. Additionally, YL-1 and E(bx), two components

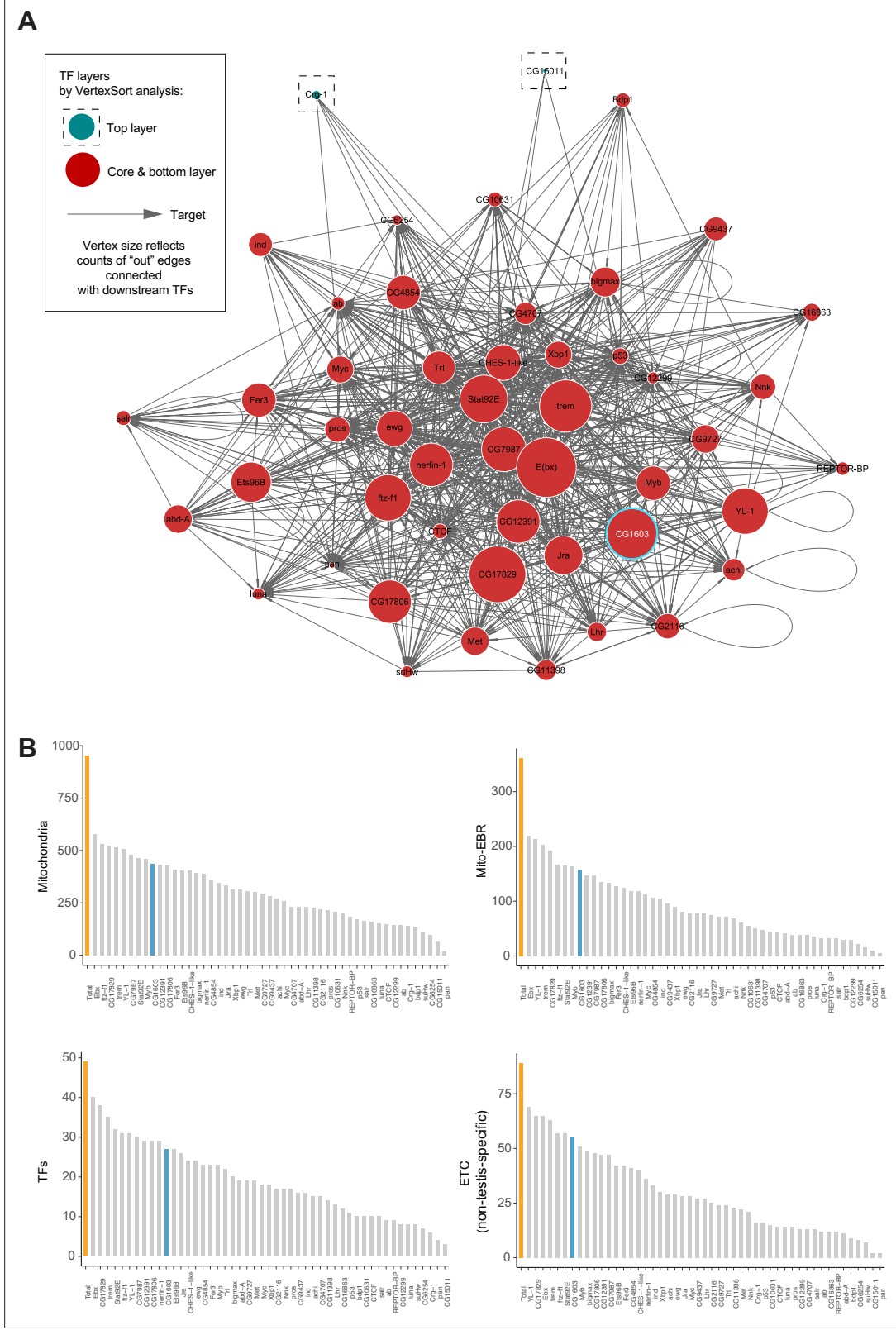

**Figure 2.** Regulatory network of mitochondrial biogenesis. (**A**) The transcriptional regulatory network of nuclear-encoded mitochondrial genes. (**B**) Bar graphs illustrating the promoter binding profiles of 49 synergistic enhancer transcription factors (TFs) within different groups of genes (nuclear-encoded mitochondrial genes, Mito-EBR genes, 49 synergistic enhancer TFs, and electron transport chain [ETC] genes). The number of genes in each group (orange) and the number of these bound by CG1603 (cyan) were highlighted.

*Figure 2 continued on next page*

*Figure 2 continued*

The online version of this article includes the following source data and figure supplement(s) for figure 2:

**Source data 1.** Raw data used to generate *Figure 2B*.

**Figure supplement 1.** The promoter binding profiles of the enhancer transcription factors (TFs) in different mitochondrial biogenesis-related groups.

**Figure supplement 1—source data 1.** Raw data used to generate *Figure 2—figure supplement 1*.

in the middle layer, are involved in chromatin remodeling (*Liang et al., 2016*; *Wysocka et al., 2006*), suggesting a potential regulation of mitochondrial biogenesis at the chromatin level.

## CG1603 regulates ETC gene expression and mitochondrial biogenesis

To validate the efficacy of this integrated genetic and bioinformatic approach, we next followed up on *CG1603*, one of the strongest hits from the primary screen (*Figure 1A and G*) and exhibited binding to a diverse array of genes associated with ETC biogenesis (*Figure 2B*, *Figure 2—figure supplement 1*, and *Supplementary file 3*). *CG1603* RNAi slightly reduced eye size. However, the combination of *CG1603* RNAi with MitoXhoI expression in the heteroplasmic background resulted in markedly smaller eyes, indicating a clear synergy between the inhibition of *CG1603* and the mtDNA deficiency. We next asked whether CG1603 was involved in mtDNA maintenance. The *Drosophila* midgut is essentially a monolayer epithelium, composed of intestine stem cells, enteroblasts, enteroendocrine cells, and enterocytes (EC). The large, flattened EC allow high-resolution imaging of mitochondria and mitochondrial nucleoids. Additionally, the simple organization and distinct cell types, containing both proliferative and terminally differentiated cells, render the midgut an ideal model to evaluate the impact of mitochondrial disruptions on cell proliferation and differentiation (*Zhang et al., 2020*). We used a 'flip-out' method to activate *CG1603* RNAi in a subset of cells (*Prober and Edgar, 2000*; *Zhang et al., 2020*), and imaged TFAM-GFP (*Zhang et al., 2016*), a marker for mitochondrial nucleoids in midgut clones. Both the total TFAM-GFP level and the number of mtDNA nucleoids (TFAM puncta) were markedly reduced in *CG1603* RNAi clones (*Figure 3A–C*), suggesting that CG1603 is necessary for maintaining the steady-state level of mtDNA. We constructed an *SDHA-mNG* reporter line by inserting the *mNeonGreen* (*mNG*) cDNA in-frame, downstream of the endogenous locus of *SDHA*, a subunit of ETC Complex II that is entirely encoded by the nuclear genome. SDHA-mNG level was notably reduced in *CG1603* RNAi clones (*Figure 3D and E*), suggesting that CG1603 is also required for the expression of nuclear-encoded ETC subunits. Different from TFAM-GFP that marks mitochondrial nucleoids and appears as puncta in mitochondria (*Chen et al., 2020*), SDHA-mNG uniformly diffused in the mitochondrial matrix (*Figure 3D*). By quantifying the total volume of SDHA-mNG positive voxels in the 3D rendering, we found that the total mitochondrial volume was also reduced in *CG1603* RNAi clones (*Figure 3F*). Collectively, these results demonstrate that CG1603 regulates the expression of genes essential for both ETC function and mitochondrial biogenesis. *CG1603* RNAi produced very few EC clones (*Figure 3A, D*, *Figure 3—figure supplement 1*), consistent with the notion that mitochondrial respiration is necessary for ISCs' differentiation (*Zhang et al., 2020*).

## CG1603 regulates cell growth and differentiation

*CG1603* encodes a C2H2 zinc finger (C2H2-ZF) protein. It has one C2H2-ZF at its N-terminus, followed by two MADF (myb/SANT-like domain in Adf-1) domains, and six additional zinc fingers at the C-terminus (*Figure 4A and B*). A PiggyBac transgene, *PBac[SAstopDSRed]LL06826*, is inserted between the exons 2 and 3 of *CG1603* locus. This modified PiggyBac mutator transgene contains splicing donors and stop codons in all three reading frames (*Schuldiner et al., 2008*), and thereby would disrupt the translation of the full-length CG1603 protein. Homozygous *PBac[SAstopDSRed]LL06826* was lethal, arrested at the second instar larval stage and eventually died after 10days (*Figure 4C*). Both the steady-state level of mtDNA and total mitochondrial mass assessed by the levels of several mitochondrial proteins were reduced in these larvae (*Figure 4D and E*), as well as the integrities and activities of ETC complexes (*Figure 4—figure supplement 1*). The lethality of this PiggyBac transgene was mapped to a genomic region spanning the *CG1603* locus (*Figure 4—figure supplement 2*). Importantly, a *P[CG1603^{gDNA}]* transgene that covers the genomic region of *CG1603* fully rescued its viability (*Figure 4A and F*). These results demonstrate that the lethality of *PBac[SAstopDSRed]LL06826* was caused by the loss of function of CG1603, and we hence named

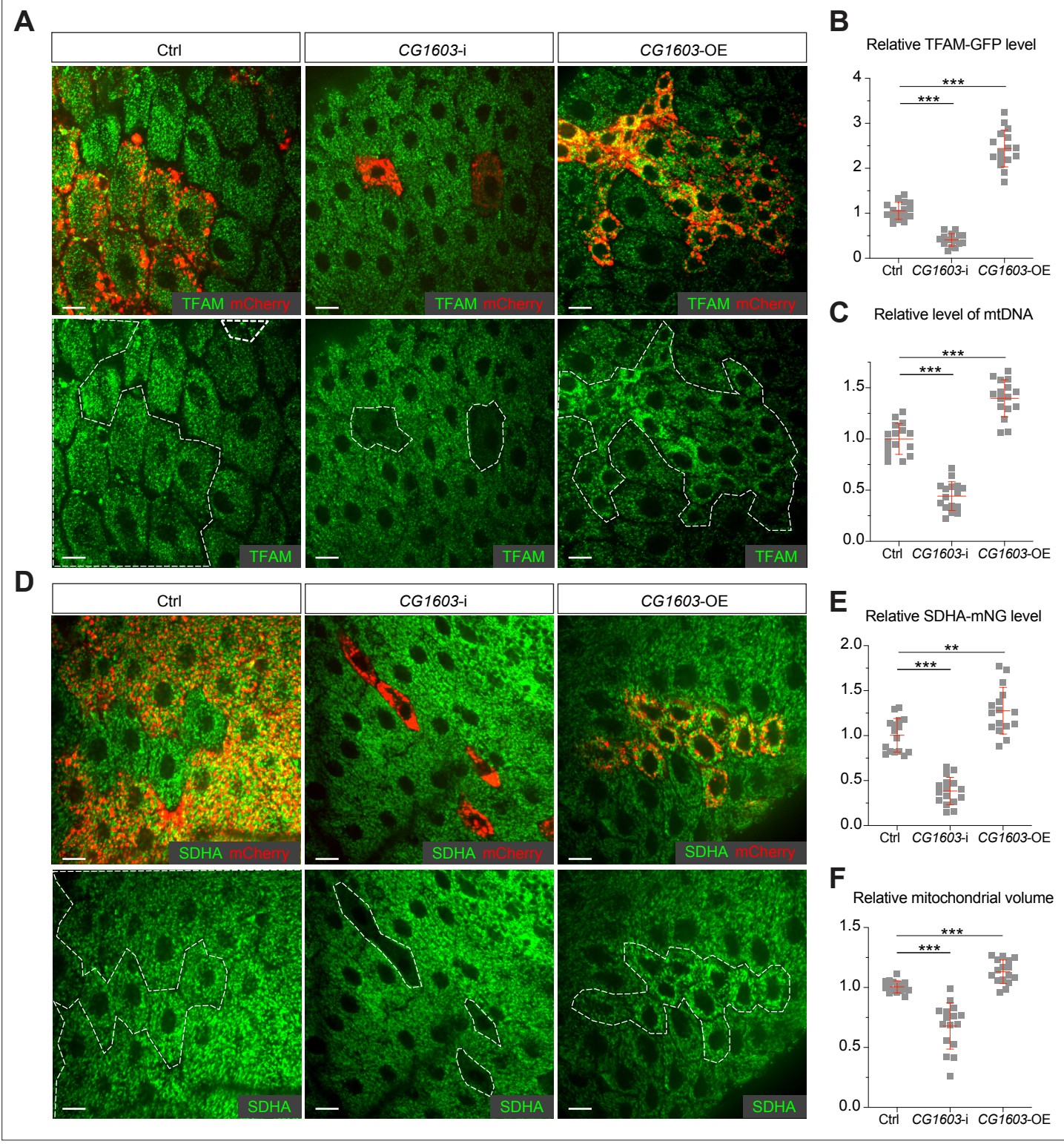

**Figure 3.** CG1603 promotes electron transport chain (ETC) gene expression and mitochondrial biogenesis. (**A, D**) Representative images of control RNAi (Ctrl), *CG1603* RNAi (*CG1603*-i), *and CG1603* overexpression (*CG1603*-OE) midgut enterocytes (EC) clones with endogenously expressed TFAM-GFP (**A**) or SDHA-mNG (**D**) visualized in green. Clones were labeled by mCherry red and compared with wild type (wt) neighbors. White dashed lines aided in illustrating clones. Scale bars: 10 µm. (**B, C, E, F**) Quantification of the relative TFAM-GFP level (**B**), the relative levels of mtDNA (**C**), the relative SDHA-mNG level (**E**), and the relative mitochondrial volume (**F**) in the EC clones to their wt neighbors. n=16 from 8 midguts for each group, error bar: SD. Two-tailed Student's *t*-test, **: p<0.01, ***: p<0.001.

*Figure 3 continued on next page*

*Figure 3 continued*

The online version of this article includes the following source data and figure supplement(s) for figure 3:

**Source data 1.** Raw data used to generate *Figure 3B, C, E, and F*.

**Figure supplement 1.** *CG1603* RNAi led to fewer enterocytes (EC) clone cells.

**Figure supplement 1—source data 1.** Raw data used to generate *Figure 3—figure supplement 1*.

it *CG1603^{PBac}* thereafter. Using FLP/FRT-mediated recombination, we generated homozygous *CG1603^{PBac}* mutant clones in both germline and follicle cells in adult ovaries. Consistent with the results of 'flip-out' RNAi experiments in the midgut, both the total TFAM level and the number of mtDNA nucleoids, visualized by an endogenously expressed TFAM-mNG reporter, were significantly reduced in *CG1603^{PBac}* clones (*Figure 5A–D*, *Figure 5—figure supplement 1A and B*). In most *CG1603^{PBac}* clones, TFAM-mNG puncta were hardly observed, demonstrating an essential role of CG1603 in mtDNA maintenance. Compared to twin clones, *CG1603^{PBac}* follicle cell clones contained significantly fewer cells, and these cells were smaller, indicating that CG1603 promotes both cell growth and cell proliferation (*Figure 5A and E*). *CG1603^{PBac}* egg chambers were also notably small, even smaller than the adjacent anterior egg chambers that are at earlier developmental stages in the same ovariole (*Figure 5A*). We assessed $\Delta\psi_m$ using the ratiometric imaging of TMRM and MitoTracker Green (*Zhang et al., 2019*). $\Delta\psi_m$ was nearly abolished in *CG1603^{PBac}* clones with reduced MitoTracker Green staining (*Figure 5F* and *Figure 5—figure supplement 1C*). All together, these observations demonstrate that CG1603 promotes mitochondrial biogenesis and is essential for ETC biogenesis.

## CG1603 is a TF regulating nuclear mitochondrial gene expression

CG1603 protein exclusively localized to the nucleus when expressed in cultured cells (*Figure 6A*). We generated a transgene expressing CG1603-mNG fusion protein by inserting *mNeonGreen* cDNA into the endogenous locus of *CG1603*. CG1603-mNG localized to nuclei in ovaries (*Figure 6B*) and directly bound to polytene chromosomes in the salivary gland (*Figure 6C*). Notably, CG1603-mNG was highly enriched on less condensed chromatin regions that had weak Hoechst staining (*Figure 6C*). We performed RNA sequencing (RNA-seq) in larvae to uncover potential targets of CG1603. Between wt and *CG1603^{PBac}* larvae, total 7635 genes were differentially expressed, including 86% nuclear-encoded mitochondrial genes (*Figure 6D* and *Supplementary file 4*; *Supplementary file 5*). Nearly half of nuclear-encoded mitochondrial genes were among 1698 genes that were reduced by more than two-fold in *CG1603* mutant (*Figure 6E* and *Supplementary file 5a*), including many structural subunit genes of all five ETC complexes (*Supplementary file 5b*), some of which were further confirmed by quantitative real-time PCR (*Figure 6F*). Gene Ontology (GO) enrichment analyses on these 1698 genes also revealed that all top 10 significantly enriched biological processes were related to mitochondria, including 'mitochondrial translation', 'mitochondrial gene expression', 'electron transport chain', 'aerobic respiration', 'cellular respiration', and 'ATP metabolic process' (*Figure 6G*).

CG1603 had 8963 binding sites (peaks) distributed over all four chromosomes (*Figure 7A* and *Supplementary file 6*). A subset of peaks showed high intensity evaluated by signalValue (*Figure 7A* and *Supplementary file 6*), which may correspond to these high-intensity CG1603-mNG bands on the polytene chromosomes of the salivary gland (*Figure 6C*). Most CG1603 binding sites (6799) were found at promoter regions, close to the transcription start site (*Figure 7B and C* and *Supplementary file 6*), which is a key feature of a typical TF. Using the RSAT 'peak-motifs' tool (*Thomas-Chollier et al., 2012*), an 8bp palindromic sequence, 'TATCGATA' emerged as the most prevalent CG1603 binding motif (*Figure 7D* and *Supplementary file 7*). CG1603 bound to the genomic regions of 50% nuclear-encoded mitochondrial genes, and among these genes, 79.5% were down-regulated in the *CG1603^{PBac}* mutant (*Figure 7E* and *Supplementary file 6*), indicating a great accordance between ChIP data and RNA-seq results. Most nuclear-encoded mitochondrial genes that were both bound by CG1603 and down-regulated in *CG1603* mutant were ETC genes or related to ETC biogenesis (*Figure 7F* and *Supplementary file 6*). Collectively, CG1603 appears to be essential for mitochondrial biogenesis and coordinates the expression of both nuclear and mtDNA genes in ETC biogenesis.

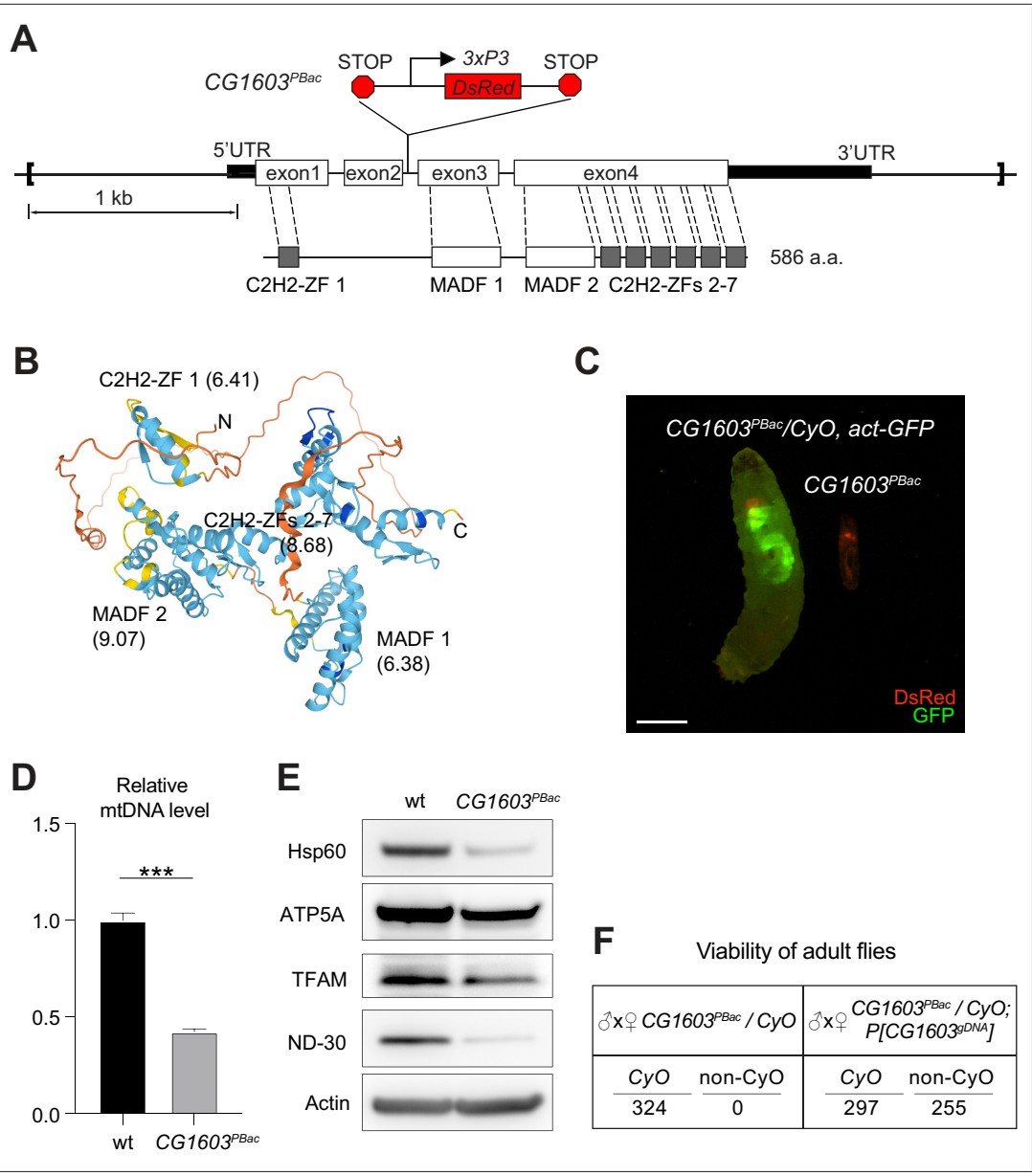

**Figure 4.** *CG1603* gene model, product, mutant, and the genomic DNA transgene. (**A**) Schematic representation of *CG1603* genomic locus, showing the *CG1603* transcript (5' and 3'UTR in black bar and four exons in white), its protein product (586 amino acids in length, and characterized by seven C2H2-ZF and two MADF domains), the *CG1603^PBac* mutant allele (with a PiggyBac insertion in the second intron, which is marked by fluorescent DsRed driven by an eye-specific *3xP3* promoter and flanked by stop codons in all three reading frames terminating translation through downstream), and the genomic region (in square brackets, from 955 bp upstream of the *CG1603* 5'UTR to 656 bp downstream of *CG1603* 3'UTR) used for the *P[CG1603^gDNA]* transgene. (**B**) Predicted 3D structure of the CG1603 protein by AlphaFold. Labels indicate the N- and C-terminus, as well as the specific protein domains along with their predicted isoelectric point (pI). (**C**) Images of *CG1603^PBac/CyO, Act-GFP*, and homozygous *CG1603^PBac* larvae cultured together at 25°C, day 4 after egg laying. Green: GFP; red: DsRed. Scale bars: 1 mm. (**D**) Relative mitochondrial DNA (mtDNA) levels in *CG1603^PBac* mutant larvae to wild type (wt) control. n=3, error bar: SD. Two-tailed Student's *t*-test, ***: p<0.001. (**E**) Western blots of mitochondrial proteins in *CG1603^PBac* mutant larvae to wt control. (**F**) *P[CG1603^gDNA]* restored viability of *CG1603^PBac* flies. The number of progenies for each genotype is listed.

The online version of this article includes the following source data and figure supplement(s) for figure 4:

**Source data 1.** Raw data used to generate *Figure 4D*.

*Figure 4 continued on next page*

*Figure 4 continued*

**Source data 2.** PDF file containing original western blots for *Figure 4E*, indicating the relevant bands and treatments.

**Source data 3.** Original files for western blot analysis displayed in *Figure 4E*.

**Figure supplement 1.** The integrities and activities of electron transport chain (ETC) complexes were reduced in *CG1603*[PBac] mutant.

**Figure supplement 1—source data 1.** PDF file containing original Blue native PAGE and in-gel activity images for *Figure 4—figure supplement 1*.

**Figure supplement 1—source data 2.** Original files for Blue native PAGE and in-gel activity analyses displayed in *Figure 4—figure supplement 1*.

**Figure supplement 2.** Adult viability phenotypes of combinations of *CG1603*[PBac] mutant, *P[CG1603*[gDNA]*]* transgene, and deficiency chromosomes.

## The integrated approach identifies YL-1 as an upstream regulator of CG1603

In the network analyses, CG1603 was positioned in the middle layer, linked to seven TFs above and six TFs below by integrating the RNA-seq result with ChIP-seq data (*Figure 8A*). Through these TFs below, CG1603 may indirectly control the expression of 2230 genes, including 291 mitochondrial genes down-regulated in *CG1603*[PBac] but not bound by CG1603 (*Figure 7E*). Using the 'flip-out' RNAi system in the midgut, we found that among seven TFs upstream of CG1603 in the network, E(bx), YL-1, trem, STAT92E, and Myb were also required for maintaining TFAM levels (*Figure 8—figure supplement 1*). To further verify their potential roles in regulating CG1603, we performed RNAi against these genes in midgut clones carrying CG1603-mNG reporter. Only *YL-1* RNAi clones displayed a marked reduction of CG1603 protein compared with neighboring cells (*Figure 8B and C*). Furthermore, overexpression of CG1603 restored the reduced eye size, TFAM-GFP, SDHA-mNG, and mtDNA levels caused by *YL-1* RNAi (*Figure 8D–J* and *Figure 8—figure supplement 2*). These results indicate that YL-1 is indeed an upstream regulator of CG1603, and through which to regulate ETC biogenesis.

## Discussion

The dual genetic control of mitochondria presents a fundamental challenge: how are the nuclear genome and mtDNA coordinated to ensure the efficiency and the integrity of oxidative phosphorylation system and other critical mitochondrial processes? In *Drosophila* ovary, the mitochondrial A-kinase-anchor-protein, MDI promotes the translation of a subset of nuclear mitochondrial proteins by cytosolic ribosomes on the mitochondrial outer membrane (*Zhang et al., 2016*). MDI's targets are predominantly ETC subunits and proteins essential for mitochondrial genome maintenance and gene expression (*Zhang et al., 2019*). This mechanism coordinates the nuclear and mitochondrial genomes to augment the ETC biogenesis that takes place in differentiating germ cells (*Wang et al., 2019*; *Wang et al., 2023*). Cytosolic and mitochondrial translation are up-regulated in concert to boost ETC biogenesis in budding yeast undergoing a metabolic shift from glycolysis to oxidative phosphorylation (*Couvillion et al., 2016*), further supporting the synchronized expression of ETC components from dual genetic origins at the translational level. Nevertheless, nuclear-encoded mitochondrial ETC subunits often exhibited a concordant expression pattern at the RNA level (*Eisen et al., 1998*), and mtDNA-encoded ETC RNAs consistently exhibited similar trends, albeit with a more gradual increase compared to their nuclear-encoded counterparts accompanying the metabolic shift (*Couvillion et al., 2016*). These observations suggest a potential coordination at the transcriptional level as well. We uncovered a zinc finger protein encoded by the *CG1603* locus as a core regulator in a transcription network regulating mitochondrial biogenesis. CG1603 promoted the expression of more than half of nuclear-encoded mitochondrial proteins, and the inhibition of CG1603 severely reduced mitochondrial mass and mtDNA contents. CG1603 targets were highly enriched in nuclear-encoded ETC subunits and essential factors required for mtDNA genome maintenance and gene expression. Thus, CG1603 not only promotes mitochondrial biogenesis in general, but also affords a transcriptional coordination of the nuclear and mitochondrial genomes in ETC biogenesis.

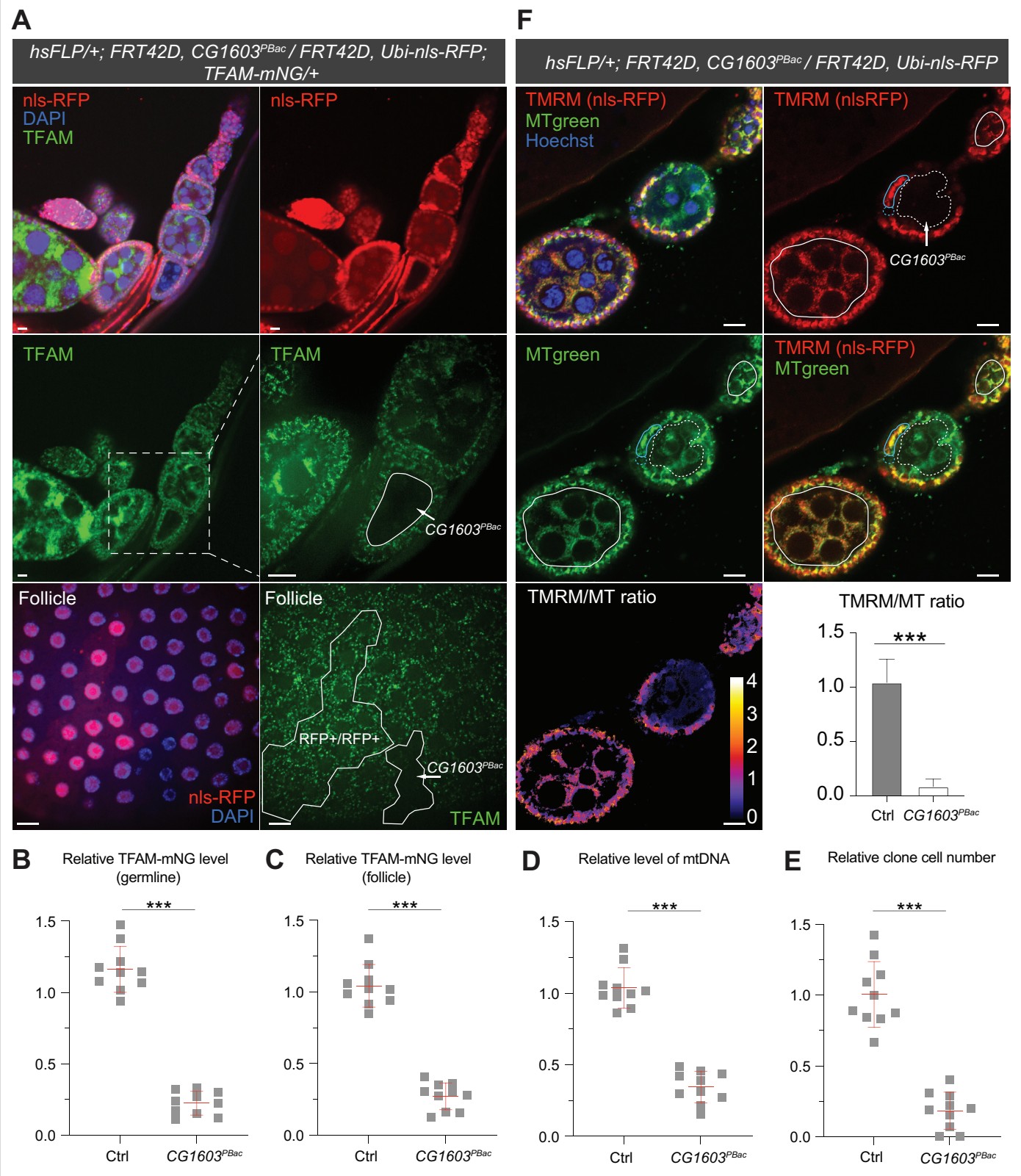

**Figure 5.** Clonal analyses confirmed CG1603's role in mitochondrial biogenesis and activity. (**A**) Representative images of *CG1603^{PBac}* mutant germline (top and middle panel) and follicle (bottom panel) clones in late-stage egg chambers of adult ovaries with endogenously expressed TFAM-mNG visualized in green. Homozygous mutant clones lacked RFP and were compared with either flanking RFP-positive cysts (germline) or homozygous wild type (wt) twin (follicle). White dashed lines aided in illustrating clones. The wt (RFP+/RFP+) follicle clone showed markedly higher RFP intensity than the

*Figure 5 continued on next page*

*Figure 5 continued*

heterozygous (RFP+/RFP-) cells, as shown in *Figure 5—figure supplement 1B*. Red: nls-RFP; blue: DAPI. Scale bars: 10 μm. (**B**) Quantification of the relative TFAM-mNG level in the homozygous *FRT42D* control and *CG1603^PBac* mutant germline clone in the early-stage egg chamber to the adjacent anterior RFP-positive cyst within the same ovariole. As shown in *Figure 5—figure supplement 1A*. n=10 for each group, error bar: SD. Two-tailed Student's *t*-test, ***: p<0.001. (**C–E**) Quantification of the relative TFAM-mNG level (**C**), the relative levels of mtDNA (**D**), and the relative clone cell number (**E**) in the homozygous *FRT42D* control and *CG1603^PBac* mutant follicle clones to their wt twins. n=10 for each group, error bar: SD. Two-tailed Student's *t*-test, ***: p<0.001. (**F**) TMRM/MitoTracker Green (MT) ratiometric live imaging and quantification of ovarioles containing homozygous *CG1603^PBac* mutant germline clones (highlighted by white dashed lines). Notably, in contrast to flanking control cysts (highlighted by white lines), $\Delta\psi_m$ was almost absent in mutant clones. Please note that compared to TMRM, nls-RFP signal was too low to be detected in ratiometric imaging. Nonetheless, the nls-RFP was readily detected in control cysts, but not in homozygous *CG1603^PBac* clones, via visual observation, as depicted in (**A**), *Figure 5—figure supplement 1A and B*. A twin pair of follicle clones in the same egg chamber were also highlighted (cyan line for control and cyan dashed line for homozygous *CG1603^PBac* mutant). The MT intensity was reduced in both the germline and follicle *CG1603^PBac* clones, compared to germ cells in adjacent egg chambers and follicle cells in the same egg chamber, respectively. Quantification with background correction for MT intensity in germline clones is shown in *Figure 5—figure supplement 1C*. Blue: Hoechst. Scale bars: 10 μm. n=8, error bar: SD. Two-tailed Student's *t*-test, ***: p<0.001.

The online version of this article includes the following source data and figure supplement(s) for figure 5:

**Source data 1.** Raw data used to generate *Figure 5B–F*.

**Figure supplement 1.** Clonal analyses confirmed CG1603's role in mitochondrial biogenesis.

**Figure supplement 1—source data 1.** Raw data used to generate *Figure 5—figure supplement 1C*.

The modifier screen in the developing eyes took advantage of the mtDNA deficiency resulted from the expression of MitoXhoI in a heteroplasmic background. Besides 77 enhancers, we also recovered 20 suppressors, of which 'RNAi+MitoXhoI' flies had larger eyes than 'RNAi-only' (*Figure 1G* and *Supplementary file 1*). Knockdown of these genes alone severely reduced eye size (*Figure 1G* and *Supplementary file 1*). Noteworthy, five of them were lethal due to the lack of head capsule that is developed from the eye antenna disc, but the viability of these RNAi flies was restored by MitoXhoI expression. Given that MitoXhoI expression also disrupts eye development, it is perplexing that the combination of RNAi and MitoXhoI expression, two genetic conditions causing the same phenotype, led to a milder phenotype. Perhaps, mtDNA deficiency caused by MitoXhoI expression triggers a retrograde signal, which boosts cellular stress responses and thereby mitigates the cell growth defects in these RNAi backgrounds.

The CG1603 belongs to a large family of C2H2-ZF TFs that contains 272 genes in the *Drosophila* genome (https://flybase.org/reports/FBgg0000732.html). It has one N-terminus C2H2-ZF, followed by two MADFs and a cluster of six C2H2-ZFs at the C-terminus (*Figure 4A*). In addition to the C2H2-ZF cluster, which predominantly mediates sequence-specific DNA binding (*Persikov et al., 2015*; *Wolfe et al., 2000*), C2H2-ZF TFs often possess additional N-terminal protein-protein interaction domains, such as KRAB, SCAN, and BTB/POZ domains in vertebrates, ZAD and BTB/POZ in *Drosophila*, for binding to transcription co-regulators (*Fedotova et al., 2017*; *Perez-Torrado et al., 2006*; *Sobocińska et al., 2021*). These interactions allow them to either activate or repress gene expression. CG1603 binds to the genomic regions of 4687 genes in the *Drosophila* genome. Among these genes, 602 and 562 genes were, respectively, decreased or increased more than two folds in *CG1603* mutant (*Supplementary file 6*). Thus, CG1603 is likely a dual-function TF, capable of both activating and repressing transcription depending on the chromosomal environment of its targets. Ying Yang 1, a well-characterized dual-function TF in mammals, contains both a transcription activation domain and a repression domain, in addition to four C2H2-ZFs at its C-terminus (*Verheul et al., 2020*). The N-terminus C2H2-ZF and the first MADF domain of CG1603 are negatively charged, and hence have low probability of binding to DNA that is also negatively charged (*Figure 4B*). In the predicted 3D structure of CG1603, the positively charged C-terminal zinc fingers and MADF-2 domain cluster in the center, while the negatively charged N-terminal C2H2-ZF and the MADF-1 extend in opposite directions (*Figure 4B*), resembling the domain arrangement of Ying Yang 1 (*Verheul et al., 2020*). MADF domains share significant similarity with Myb/SANT domains that may bind to either DNA or proteins (*Maheshwari et al., 2008*). Some MADF domains, due to their negative charge, have been proposed to interact with positively charged histone tails, similar to the Myb/SANT domain in a well-known chromatin remodeler ISWI, suggesting a potential role in chromatin remodeling (*Maheshwari et al., 2008*). Notably, some chromatin remodelers possess tandem Myb/SANT domains that can directly

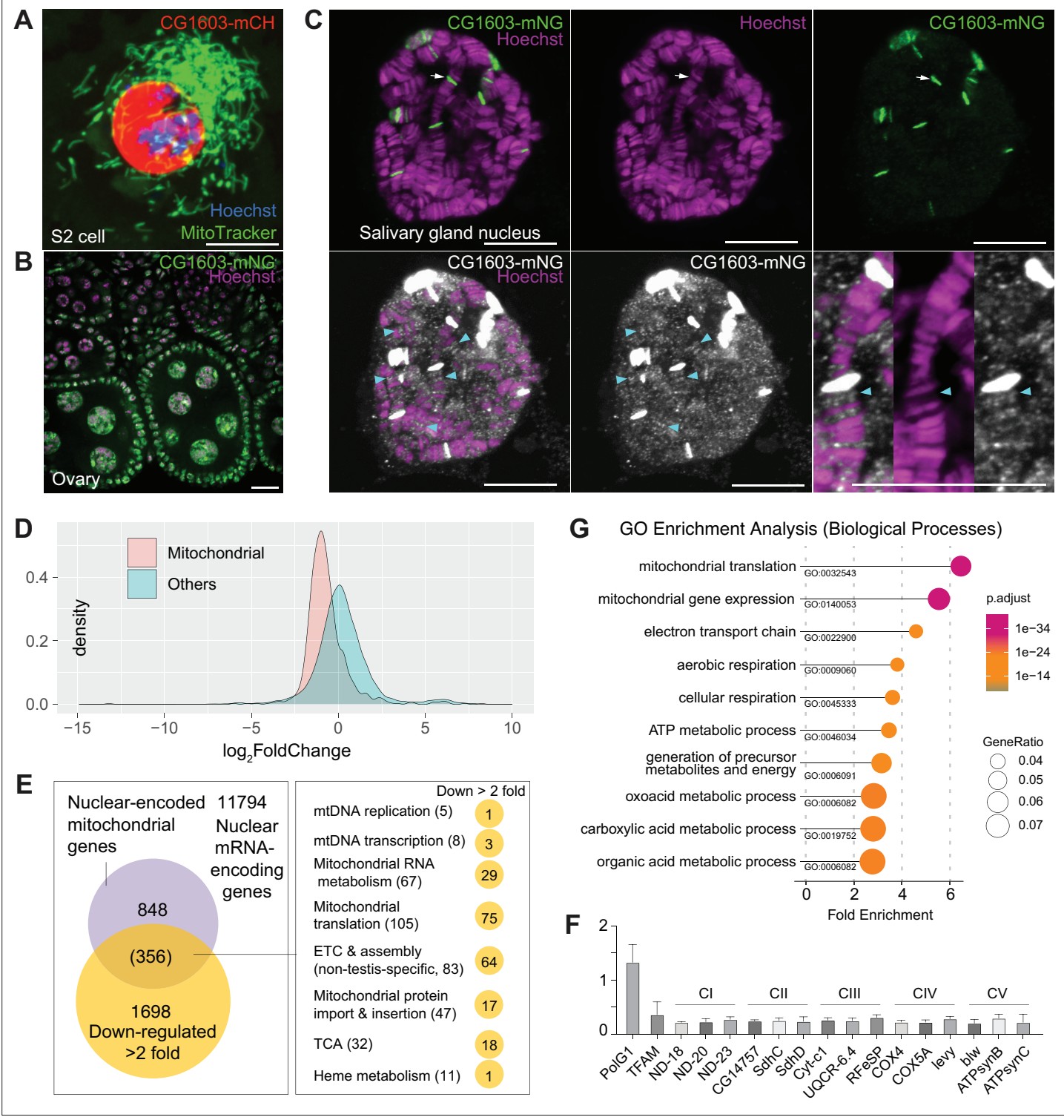

**Figure 6.** CG1603 localizes in the nucleus and is essential for regulating nuclear mitochondrial gene expression. (**A–B**) Representative images showing the nuclear localization of CG1603 protein in cultured S2 cells (**A**) and adult ovary (**B**). Green: MitoTracker Green in S2 cell, and CG1603-mNG in tissues; red: CG1603-mCH; blue and magenta: Hoechst. Scale bars: 10 μm. (**C**) Representative images showing bindings of endogenously expressed CG1603 proteins to less condensed euchromatin regions in the polytene chromosomes of a salivary gland. High-intensity CG1603-mNG bands were visualized in green in the upper panel and indicated by arrows, and low-intensity bands were pesudo-colored in white in theblower panel and indicated by arrow heads. The images of the lower panel were same as these in the upper panel, but digitally enhanced. Magenta: Hoechst. Scale bars: 10 μm. (**D**) Density plot illustrating the distribution of expression changes of the nuclear-encoded mitochondrial and non-mitochondrial genes in *CG1603^PBac* mutant.

*Figure 6 continued*

(**E**) Graph illustrating the overlap between nuclear-encoded mitochondrial genes and differentially expressed genes (DEGs) that down-regulated >2-fold, as well as the distribution of the overlapped genes in different mitochondrial function categories. (**F**) Relative mRNA levels of several electron transport chain (ETC) biogenesis-related genes in *CG1603^PBac^* mutant larvae to control, measured by real-time PCR. n=3, error bar: SD. (**G**) Gene Ontology (GO) enrichment analyses of DEGs that down-regulated >2-fold. The top 10 enriched biological processes are shown.

The online version of this article includes the following source data for figure 6:

**Source data 1.** Raw data used to generate *Figure 6D, F, and G*.

---

interact with histones and DNA or histone remodeling enzymes like ISWI and SMRT (***Boyer et al., 2004***). It is plausible that CG1603 may play a role in chromatin remodeling directly or by recruiting nucleosome remodeling factors to its binding sites, thereby modulating gene transcription in those regions.

Notably, CG1603 had no impact on the expression of one-third of its binding genes (***Supplementary file 6***), highlighting that DNA binding profiling alone is not sufficient to predict the function of a TF. Nonetheless, the network analyses on TFs' ChIP-seq data allowed us to construct a potential regulatory network among these TFs, which subsequently served as a blueprint for genetic analyses to verify potential regulations. Total seven TFs were upstream of CG1603 in the network and emerged as positive hits in the initial screen in the eye. RNAi against five of them led to reduced TFAM levels in the midgut, while the other two had no noticeable phenotype, suggesting that these two TFs may regulate mitochondrial biogenesis in a tissue-specific manner. Only YL-1 was confirmed to act upstream of CG1603 based on the genetic epistasis analysis, further indicating the necessity of combining genomic, bioinformatic, and genetic analyses to gain more reliable and comprehensive understanding on transcriptional regulations. YL-1 is one of DNA binding subunits of the SRCAP complex, which is essential for histone H2A.Z incorporation and replacement (***Liang et al., 2016***). Recently, it has been shown that both SRCAP complex and H2A.Z are necessary for the transcription of nuclear-encoded mitochondrial genes (***Lowden et al., 2021***; ***Xu et al., 2021***). Our work offers a mechanistic insight into how CG1603 and its upstream regulator, YL-1, may regulate mitochondrial biogenesis at nucleosome and chromatin levels. Currently, most known transcription paradigms controlling mitochondrial biogenesis are centered on TFs and co-activators that stabilize or directly stimulate the core transcription machinery. The YL-1 to CG1603 cascade may represent a previously underappreciated layer of transcriptional regulation on ETC biogenesis and could act in concert with NRF1 and other TFs to coordinate both the nuclear and mitochondrial genomes in ETC biogenesis.

# Materials and methods

## Key resources table

| Reagent type (species) or resource | Designation | Source or reference | Identifiers | Additional information |
|---|---|---|---|---|
| Gene (*Drosophila melanogaster*) | *CG1603* | GenBank | FLYB:FBgn0033185 | |
| Genetic reagent (*D. melanogaster*) | *w^1118^* | Bloomington Drosophila Stock Center | BDSC: 3605; RRID:BDSC_3605 | |
| Genetic reagent (*D. melanogaster*) | Heteroplasmic *Sco/CyO, UAS-mitoXhoI; eyeless-GAL4* | This paper | | See Materials and methods, Section Fly genetics |
| Genetic reagent (*D. melanogaster*) | RNAi stains used for genetic screen | Bloomington Drosophila Stock Center; Vienna *Drosophila* Resource Center | See 'ID', 'SYMBOL', and 'Stock #' column in *Supplementary file 1* | Stock # begins with 'v' is from VDRC, otherwise BDSC |
| Genetic reagent (*D. melanogaster*) | *UAS-Luciferase* | Bloomington Drosophila Stock Center | BDSC:35788 | |
| Genetic reagent (*D. melanogaster*) | *TFAM-GFP* | PMID:27053724 | | |

*Continued on next page*

*Continued*

| Reagent type (species) or resource | Designation | Source or reference | Identifiers | Additional information |
|---|---|---|---|---|
| Genetic reagent (*D. melanogaster*) | *hsFLP* | Bloomington Drosophila Stock Center | BDSC:7 | |
| Genetic reagent (*D. melanogaster*) | *Act >CD2>GAL4, UAS-mCD8::mCherry* | This paper | | See Materials and methods, Section Fly genetics |
| Genetic reagent (*D. melanogaster*) | *PBac[SAstopDsRed]LL06826* | Kyoto *Drosophila* Stock Center | Kyoto:141919 | |
| Genetic reagent (*D. melanogaster*) | *CyO, act-GFP* | Bloomington *Drosophila* Stock Center | BDSC:4533 | |
| Genetic reagent (*D. melanogaster*) | *FRT42D* | Bloomington *Drosophila* Stock Center | BDSC:1802 | |
| Genetic reagent (*D. melanogaster*) | *FRT42D, Ubi-nls-RFP* | Bloomington *Drosophila* Stock Center | BDSC:35496 | |
| Genetic reagent (*D. melanogaster*) | *Def*$^{k08815}$ | Bloomington *Drosophila* Stock Center | BDSC:10818 | |
| Genetic reagent (*D. melanogaster*) | *Def*$^{Exel6052}$ | Bloomington *Drosophila* Stock Center | BDSC:7534 | |
| Genetic reagent (*D. melanogaster*) | *Def*$^{Exel6053}$ | Bloomington *Drosophila* Stock Center | BDSC:7535 | |
| Genetic reagent (*D. melanogaster*) | *UASz-CG1603* | This paper | | See Materials and methods, Section Transgenic flies |
| Genetic reagent (*D. melanogaster*) | *P[CG1603*$^{gDNA}$*]* | This paper | | See Materials and methods, Section Transgenic flies |
| Genetic reagent (*D. melanogaster*) | *SDHA-mNeonGreen* | This paper | | See Materials and methods, Section Transgenic flies |
| Genetic reagent (*D. melanogaster*) | *TFAM-mNeonGreen* | This paper | | See Materials and methods, Section Transgenic flies |
| Genetic reagent (*D. melanogaster*) | *CG1603-Halo-mNeonGreen* | This paper | | See Materials and methods, Section Transgenic flies |
| Cell line (*D. melanogaster*) | S2 | *Drosophila* Genomics Resource Center | FLYB:FBtc0000181; RRID:CVCL_Z992 | |
| Antibody | Anti-Actin (Mouse monoclonal) | MilliporeSigma | Cat# MAB1501; RRID:AB_2223041 | WB (1:1000) |
| Antibody | Anti-ATP5A (Mouse monoclonal) | abcam | Cat# 15H4C4; RRID:AB_301447 | WB (1:2000) |
| Antibody | Anti-ND30 (Mouse monoclonal) | abcam | Cat# 17D95; | WB (1:1000) |
| Antibody | Anti-TFAM (Rabbit polyclonal) | PMID:35449456 | | WB (1:1000) |
| Antibody | Anti-HSP60 (Rabbit polyclonal) | Cell Signaling | Cat# 4870; RRID:AB_2295614 | WB (1:1000) |
| Antibody | Anti-rabbit IgG, HRP-linked (Goat polyclonal) | Cell Signaling | Cat# 7074; RRID:AB_2099233 | WB (1:2000) |
| Antibody | Anti-mouse IgG, HRP-linked (Horse polyclonal) | Cell Signaling | Cat# 7076; RRID:AB_330924 | WB (1:2000) |

*Continued on next page*

*Continued*

| Reagent type (species) or resource | Designation | Source or reference | Identifiers | Additional information |
|---|---|---|---|---|
| Recombinant DNA reagent | pIB-CG1603-mCherry (plasmid) | This paper | | See Materials and methods, Section Cell culture and gene expression |
| Recombinant DNA reagent | pIB/V5-His (plasmid) | Thermo Fisher Scientific | Cat# V802001 | |
| Sequence-based reagent | Primers for real-time PCR | This paper | | Listed in *Supplementary file 8* |
| Commercial assay or kit | Effectene Transfection Reagent | QIAGEN | Cat# 301425 | |
| Commercial assay or kit | NativePAGE Sample Prep Kit | Thermo Fisher Scientific | Cat# BN2008 | |
| Commercial assay or kit | Pierce BCA Protein Assay Kit | Thermo Fisher Scientific | Cat# 23225 | |
| Commercial assay or kit | NativePAGE Running Buffer Kit | Thermo Fisher Scientific | Cat# BN2007 | |
| Commercial assay or kit | DNeasy Blood & Tissue Kit | QIAGEN | Cat# 69504 | |
| Commercial assay or kit | RNeasy Mini Kit | QIAGEN | Cat# 74104 | |
| Commercial assay or kit | SuperScript VILO cDNA Synthesis Kit | Thermo Fisher Scientific | Cat# 11754050 | |
| Commercial assay or kit | PowerTrack SYBR Green Master Mix | Thermo Fisher Scientific | Cat# A46012 | |
| Chemical compound, drug | TRIzol | Thermo Fisher Scientific | Cat# 15596026 | |
| Chemical compound, drug | DAPI | Thermo Fisher Scientific | Cat# D1306 | |
| Chemical compound, drug | Hoechst 33342 | Thermo Fisher Scientific | Cat# H1399 | |
| Chemical compound, drug | TMRM | Thermo Fisher Scientific | Cat# I34361 | |
| Chemical compound, drug | MitoTracker Green | Thermo Fisher Scientific | Cat# M7514 | |
| Software, algorithm | Imaris | Oxford Instruments | | See Materials and methods, Section Imaging analyses |
| Software, algorithm | Fiji/ImageJ | NIH | | See Materials and methods, Section Imaging analyses |
| Software, algorithm | FastQC | Babraham Bioinformatics | | See Materials and methods, Section RNA-seq analysis |
| Software, algorithm | STAR | PMID:23104886 | | See Materials and methods, Section RNA-seq analysis |
| Software, algorithm | HTseq | PMID:35311944 | | See Materials and methods, Section RNA-seq analysis |

*Continued on next page*

*Continued*

| Reagent type (species) or resource | Designation | Source or reference | Identifiers | Additional information |
|---|---|---|---|---|
| Software, algorithm | DESeq2 | PMID:25516281 | | See Materials and methods, Section RNA-seq analysis |
| Software, algorithm | clusterProfiler | PMID:22455463 | | See Materials and methods, Section RNA-seq analysis |
| Software, algorithm | BWA | PMID:19451168 | | See Materials and methods, Section ChIP-seq computational analysis |
| Software, algorithm | Samtools | PMID:19505943 | | See Materials and methods, Section ChIP-seq computational analysis |
| Software, algorithm | MACS2 | PMID:18798982 | | See Materials and methods, Section ChIP-seq computational analysis |
| Software, algorithm | ChIPseeker | PMID:25765347 | | See Materials and methods, Section ChIP-seq computational analysis |
| Software, algorithm | VertexSort | PMID:19690563 | | See Materials and methods, Section ChIP-seq computational analysis |
| Software, algorithm | RSAT | PMID:22836136 | | See Materials and methods, Section ChIP-seq computational analysis |
| Software, algorithm | SMART | PMID:10592234 | | See Materials and methods, Section Prediction of protein domains, isoelectric point, net charge, and structure |
| Software, algorithm | AlphaFold | PMID:34265844 | | See Materials and methods, Section Prediction of protein domains, isoelectric point, net charge, and structure |

## Fly genetics

Flies were maintained on standard cornmeal medium at 25°C, unless otherwise stated. Heteroplasmic lines that contain ~50% XhoI-resistant *mt:CoI^T300I* genome (*Hill et al., 2014*) were maintained at 18°C. The heteroplasmic *w^1118^; Sco/CyO, UAS-mitoXhoI; eyeless-GAL4* females were crossed with different RNAi lines to generate male offspring for assessing adult eye morphology. RNAi lines used in the screen were obtained from the Bloomington *Drosophila* Stock Center (BDSC), or Vienna Drosophila Resource Center, and listed in *Supplementary file 1*. *UAS-Luciferase* (BDSC#35788) was used as the transgene control. *TFAM-GFP* reporter line was described previously (*Zhang et al., 2016*). *Act >CD2>GAL4, UAS-mCD8::mCherry,* and *hsFLP* (BDSC#7) were used to generate 'flip-out' clones in midguts. We found that the leakage expression of flippase at 22°C was sufficient to induce 'flip-out' clones. *PBac[SAstopDsRed]LL06826* (Kyoto#141919) was obtained from Kyoto Drosophila Stock Center, and backcrossed to *w^1118^* for six generations before phenotypic analyses. A fluorescent '*CyO, act-GFP*' (BDSC#4533) was used for selecting homozygous mutant larvae. *PBac[SAstopDsRed] LL06826* was recombined with *FRT42D* (BDSC #1802) to generate *FRT42D, CG1603^PBac^*, which was

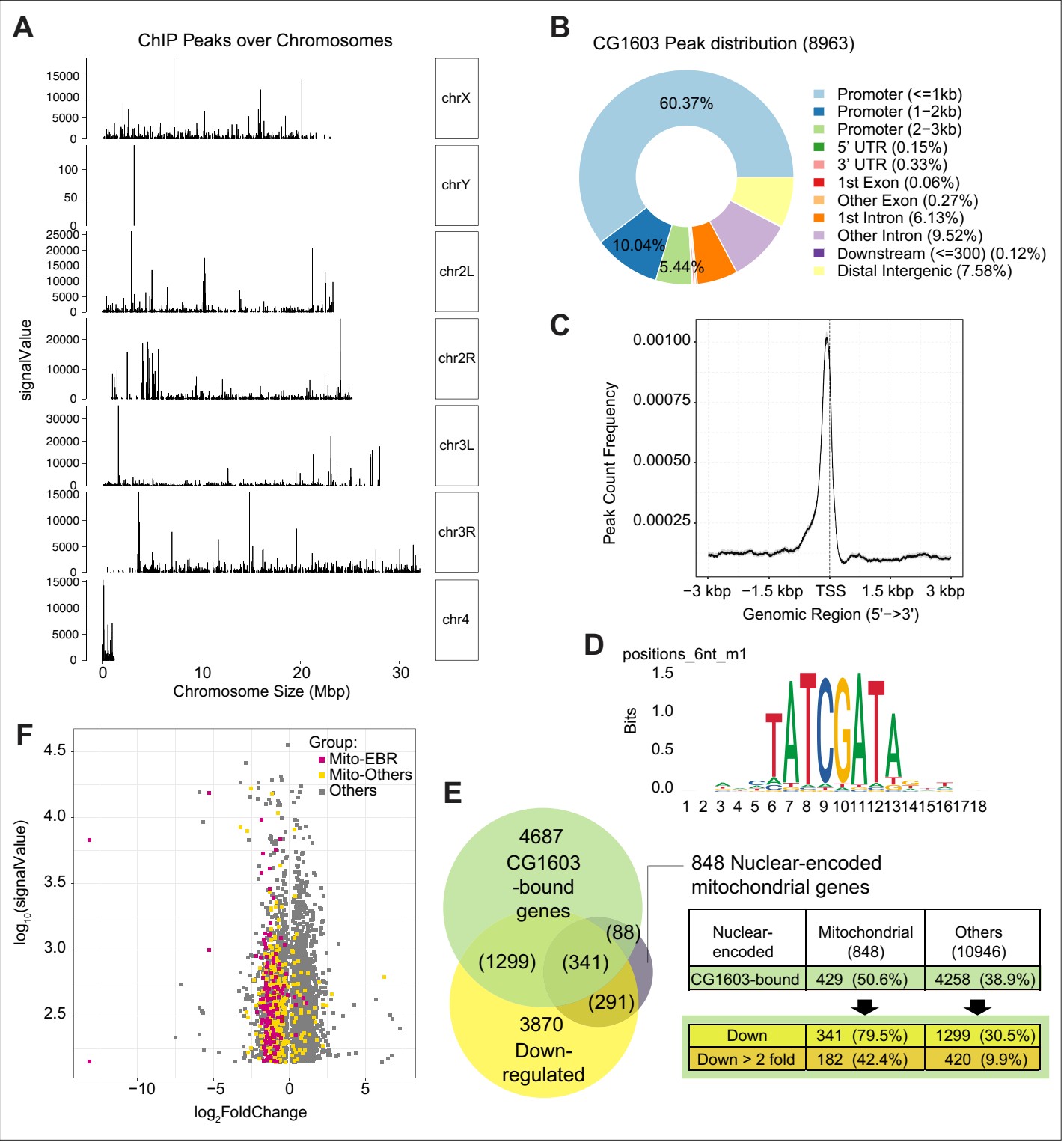

**Figure 7.** ChIP analysis identified nuclear mitochondrial genes that may be directly regulated by CG1603. (**A**) CG1603 ChIP peaks over all chromosomes. (**B**) Genomic distribution of CG1603 peaks. (**C**) Average profile of CG1603 peaks binding to transcription start site (TSS) regions. (**D**) Representative binding motif discovered with CG1603 ChIP peaks. (**E**) The number of nuclear-encoded mitochondrial and non-mitochondrial genes bound by CG1603, and the overlapping down-regulated differentially expressed genes (DEGs) in each group. (**F**) Scatterplot illustrating the signalValue of CG1603 ChIP peaks (y-axis) and log2 fold change in expression of DEGs between *CG1603*[*PBac*] mutant and control (x-axis). Mito-EBR: genes related to electron transport chain (ETC) biogenesis and maintenance, including ETC subunits and assembly factors, mitochondrial DNA (mtDNA) replication and transcription, mitochondrial RNA metabolism and translation, as well as mitochondrial protein import and membrane insertion machinery.

*Figure 7 continued on next page*

*Figure 7 continued*

The online version of this article includes the following source data for figure 7:

**Source data 1.** Raw data used to generate *Figure 7A–D and F*.

crossed with *hs-flp; FRT42D, Ubi-nls-RFP* (derived from BDSC#35496) for generating mitotic clones in ovaries (*Laws and Drummond-Barbosa, 2015*). Briefly, 0- to 2-day-old females were transferred along with sibling males to the Kimwipe-semi-covered vials, then passed to 37°C water bath, heat shocked for 1 hr, twice daily, for 3 consecutive days. The clones were assessed 7–10days after the final heat shock. *Def^{k08815}*(BDSC#10818), *Def^{Exel6052}*(BDSC#7534), and *Def^{Exel6053}*(BDSC#7535) were obtained from BDSC.

## Cell culture and gene expression

S2 cells from Drosophila Genomics Resource Center (DGRC) were cultured as previously described *Zhang et al., 2015* following the online instruction (DRSC, https://fgr.hms.harvard.edu/fly-cell-culture). Briefly, cells were maintained in Schneider's medium (Thermo Fisher Scientific) with 10% heat inactivated fetal bovine serum (FBS, Thermo Fisher Scientific) and 1% penicillin-streptomycin (Thermo Fisher Scientific) at 27°C. Effectene Transfection Reagent (QIAGEN) was used for plasmids transfection following the manufacturer's instructions. For expression in S2 cells, the coding sequence of *CG1603* was cloned into a pIB vector (Thermo Fisher Scientific), with an *mCherry* coding sequence fused at the 3' end.

## Transgenic flies

*UASz-CG1603* plasmid was generated by inserting *CG1603* coding sequence between the Acc65I and XbaI sites of pUASz1.0 (https://dgrc.bio.indiana.edu//stock/1431; RRID:DGRC_1431). *UASz-CG1603* was inserted into either attP2 or attP40 (Bestgene Inc) using PhiC31 integrase-mediated site-specific transformation, to generate transgenes on third and second chromosome, respectively.

The DNA fragment spanning *CG1603* genomic region was amplified by PCR and subcloned into a pattB vector (https://dgrc.bio.indiana.edu//stock/1420; RRID:DGRC_1420). The resulted plasmid was inserted into attP2 site (Bestgene Inc) using PhiC31 integrase-mediated site-specific transformation to generate the transgene *P[CG1603^{gDNA}]*.

*SDHA-mNeonGreen* reporter line was generated using a previously published method (*Wang et al., 2019*). The targeting cassette comprising of 1kb genomic DNA fragment upstream of *SDHA* stop codon, *mNeonGreen* coding sequence, a fragment containing *GMR-Hid* flanked by two FRT sites, and 1kb genomic DNA fragment downstream of *SDHA* stop codon was inserted into a pENTR vector to make the homology donor construct. This donor construct and an *SDHA* chiRNA construct (*gRNA-SDHA* recognizes GTAGACATCCGTACGAGTGA[TGG]) were injected into the *Vasa-Cas9* expressing embryos (BDSC#51323). G0 adults were crossed with *w^{1118}* files (BDSC#3605), and progeny with small-eye phenotype were selected as candidates due to the expression of *GMR-Hid*. To remove the *GMR-Hid* cassette, the *SDHA-mNeon-GMR-Hid* flies were crossed with *nos-Gal4; UASp-FLP*. The F1 progeny with the genotype of *nos-Gal4/SDHA-mNeon-GMR-Hid; UASp-FLP/+* were selected and crossed with *Sco/CyO*. The F2 flies of *SDHA-mNeon/CyO* with normal white eyes were selected and maintained.

For *TFAM-mNeonGreen* and *CG1603-Halo-mNeonGreen* knock-In lines, the targeting cassette comprising of 1kb genomic DNA fragment upstream of the stop codon, either *mNeonGreen* or *Halo-mNeonGreen* coding sequence, and 1kb genomic DNA fragment downstream of the stop codon was inserted into pOT2 vector to generate the donor constructs. Each donor construct and the corresponding chiRNA construct (*gRNA for TFAM*: ATGATTTGTGAATTATGTGATGG; *gRNA for CG1603*: GGAATGAACTCTCGCCTTGAGGG) were injected into Vasa-Cas9 expressing embryos (BDSC#51323 or BDSC#51324). G0 adults were crossed with w1118 files, and the progeny carrying the *mNeonGreen* insertions were screened by PCR. Primers for TFAM-mNeonGreen are GCTCGCTGATCAACAA AGTC and GGTGGACTTCAGGTTTAACTCC. Primers for CG1603-mNeonGreen are AGTGCGAG TTCCTCAGT-GTG and CGCCCAGGACTTCCACATAA.

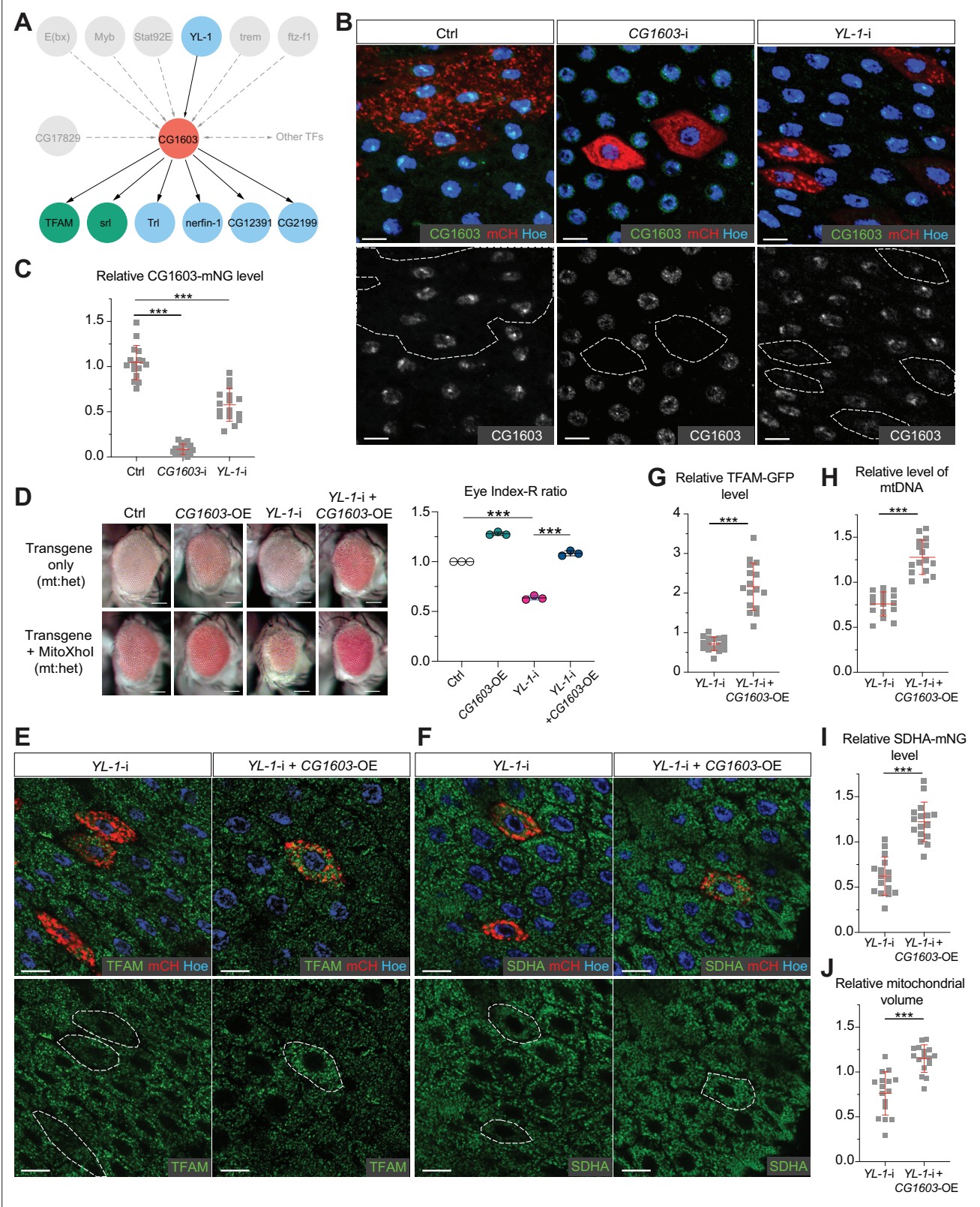

**Figure 8.** YL-1 is an upstream regulator of CG1603. (**A**) Schematic graph illustrating the CG1603 upstream and downstream (co-)TFs (transcription factors) involved in regulating mitochondrial electron transport chain (ETC) biogenesis, inferred from ChIP-seq, RNA-seq, and genetics data. (**B**) Representative images of control RNAi (Ctrl), *CG1603* RNAi (*CG1603*-i), and *YL-1* RNAi (*YL-1*-i) midgut enterocytes (EC) clones with endogenously expressed CG1603-mNG visualized in green or white. Clones were labeled by mCherry red and compared with wild type (wt) neighbors. White dashed

*Figure 8 continued on next page*

*Figure 8 continued*

lines aided in illustrating clones. Blue: Hoechst. Scale bars: 10 µm. (**C**) Quantification of the relative CG1603-mNG level in the EC clones to their wt neighbors. n=16 from 8 midguts for each group, error bar: SD. Two-tailed Student's *t*-test, \*\*\*: p<0.001. (**D**) Representative eye image and Index-R ratio (RNAi+mitoXhoI/RNAi-only) of adult flies with indicated genotypes. Three biological repeats were performed for each group, error bar: SD. Two-tailed Student's *t*-test, \*\*\*: p<0.001. Scale bars: 100 µm. (**E–F**) Representative images of *YL-1* RNAi (*YL-1*-i) and *YL-1* RNAi+*CG1603* overexpression (*YL-1*-i+*CG1603* OE) midgut EC clones with endogenously expressed TFAM-GFP (**E**) or SDHA-mNG (**F**) visualized in green. Clones were labeled by mCherry red and compared with wt neighbors. Blue: Hoechst. Scale bars: 10 µm. (**G–J**) Quantification of the relative TFAM-GFP level (**G**), the relative levels of mtDNA (**H**), the relative SDHA-mNG level (**I**), and the relative mitochondrial volume (**J**) in the EC clones to their wt neighbors. n=16 from 8 midguts for each group, error bar: SD. Two-tailed Student's *t*-test, \*\*\*: p<0.001.

The online version of this article includes the following source data and figure supplement(s) for figure 8:

**Source data 1.** Raw data used to generate *Figure 8C, D, and G–J*.

**Figure supplement 1.** TFAM levels in the RNAi midgut clones of transcription factors (TFs) upstream of CG1603 in the network.

**Figure supplement 1—source data 1.** Raw data used to generate *Figure 8—figure supplement 1B*.

**Figure supplement 2.** Overexpression of CG1603 restored the reduced mitochondrial DNA (mtDNA) level caused by *YL-1* RNAi in the eye discs.

**Figure supplement 2—source data 1.** Raw data used to generate *Figure 8—figure supplement 2*.

## RNA-seq analysis

For bulk RNA-seq analysis, total RNA was extracted from wt and CG1603 mutant larvae (48hr after egg laying) by TRIzol (Thermo Fisher Scientific) following the standard protocol. Three samples were used for each genotype. Poly (A) capture libraries were generated at the DNA Sequencing and Genomics Core, NHLBI, NIH. RNA-seq was performed with using an NovaSeq 6000 (Illumina) and 100bp paired-end reads were generated at the DNA Sequencing and Genomics Core, NHLBI, NIH. Sequencing data were analyzed following the Bioinformatics Pipeline of mRNA Analysis, NCI, NIH. After quality assessment of FASTQ files using FastQC (https://www.bioinformatics.babraham.ac.uk/projects/fastqc), paired-end reads were aligned against *D. melanogaster* reference genome (Dmel6) using a two-pass method with STAR (v2.7.9a) (*Dobin et al., 2013*). Gene-level read counts were produced by HTseq (v0.11.4) (*Putri et al., 2022*). Differential expression analysis at the gene level was carried out using DESeq2 open-source R package (*Love et al., 2014*) with an FDR cut-off of 0.05. GO enrichment analysis was performed using clusterProfiler R package (*Yu et al., 2012*) with the log2 fold change cut-off>1and <–1 for up-regulated and down-regulated genes, respectively. A density plot was generated by ggplot2 R package (https://ggplot2.tidyverse.org). *Drosophila* mitochondrial genes and subgroups were referenced against a modified MitoCarta 3.0 inventory (*Rath et al., 2021*; *Wang et al., 2019*).

## ChIP-seq computational analysis

ChIP-seq reads in FASTQ format and the narrowPeak output files for each candidate TF were downloaded from ENCODE Project open resource (https://www.encodeproject.org; *Kudron et al., 2018*). ChIP-seq reads were aligned to the *D. melanogaster* reference genome (Dmel6) using BWA (v0.7.17) (*Li and Durbin, 2009*). SAM files were sorted and compressed into BAM format with Samtools (v1.16.1) (*Li et al., 2009*). Replicates were merged by Picard tools (v2.27.3, https://broadinstitute.github.io/picard) using lenient criteria, and all alignments with an MAPQ value less than 20 were removed. Lags prediction and peak-calling were done with MACS2 (v2.2.7.1) *Zhang et al., 2008* following the ENCODE TF ChIP pipeline with IDR analysis performed for consistency analysis (https://github.com/mforde84/ENCODE_TF_ChIP_pipeline; *mforde84, 2016*). Peak annotation and analysis of profile of ChIP peaks binding to TSS regions were performed with ChIPseeker R package (*Yu et al., 2015*). Transcription network was analyzed and visualized with VertexSort (*Jothi et al., 2009*) and igraph (https://igraph.org) R packages, respectively, and ChIP peaks of each TF identified in the gene promoter regions (<2kb) were used for analyses. CG1603 binding motif discovery was done using online integrated pipeline 'peak-motifs' of RSAT tools (https://rsat.france-bioinformatique.fr/rsat/peak-motifs_form.cgi; *Thomas-Chollier et al., 2012*).

## Imaging analyses

Imaging analyses were performed as previously described (*Zhang et al., 2020*) using the Zeiss Axio Observer equipped with a Perkin Elmer spinning disk confocal system or a Zeiss LSM880 confocal

system. Tissues were dissected out and rinsed in room temperature Schneider's medium (Thermo Fisher Scientific) supplied with 10% heat inactivated FBS (Thermo Fisher Scientific), and then used for either direct imaging or further staining and fixation. For live imaging, a Zeiss incubation system was used to maintain proper temperature and humidity. Live tissues were mounted with medium on the coverslip in a custom-made metal frame and then covered with a small piece of Saran wrap before imaging. For tissue fixation, PBS containing 4% PFA was used, followed by three washes with PBS. Hoechst 33342 and DAPI (5µg/ml, Thermo Fisher Scientific) incubation in PBS for 5 min was used for nuclear staining of live tissues and fixed tissues, respectively. The image processing and quantification were performed by Volocity (Perkin Elmer, for image acquisition), Zen (Zeiss, for image acquisition), Imaris (Oxford Instruments, https://imaris.oxinst.com/, for 3D surface, voxels, and intensity statistics), and Fiji/ImageJ software (NIH, https://fiji.sc/, for image processing and statistics) based on the previously published methods (*Liu et al., 2022*; *Wang et al., 2023*). The relative level of TFAM-GFP or TFAM-mNeonGreen, or relative SDHA-mNeonGreen level, was calculated as the ratio of the mean fluorescence intensity in the RNAi or mutant clone to that of its neighboring control, with background correction. The relative CG1603-mNG level was calculated as the ratio of the mean nuclear fluorescence intensity in the RNAi clone to that of its neighboring control, with background correction. The relative level of mtDNA was determined by calculating the ratio of the TFAM-GFP or TFAM-mNeonGreen puncta number in the RNAi or mutant clone, standardized by clone volume, to that of its neighboring control. The relative mitochondrial volume was calculated as the ratio of the total SDHA-GFP positive voxels with local contrast in the RNAi clone, standardized by clone volume, to that of its neighboring control.

Mitochondrial membrane potential was detected using a protocol adopted from a previous study (*Zhang et al., 2020*; *Zhang et al., 2019*). Briefly, after dissection, adult ovaries were incubated in the Schneider's medium containing TMRM (200nM, Thermo Fisher Scientific) and MitoTracker Green (200nM, Thermo Fisher Scientific) for 20min at room temperature, rinsed with PBS for three times, and then imaged live within 1hr. TMRM and MitoTracker signal intensities were quantified and ratiometric images were generated using Fiji/ImageJ software (NIH). Mitochondrial membrane potential was computed as the ratio of the mean intensity of TMRM to MitoTracker fluorescence with background correction.

## Western blot

Protein extracts from wt and *CG1603* mutant larvae tissues (48hr after egg laying) were prepared using the RIPA buffer (MilliporeSigma) with Halt Protease Inhibitor Cocktail (Thermo Fisher Scientific), 5mM NaF (MilliporeSigma), and 1mM $Na_3VO_4$ (MilliporeSigma). Western blot was performed using a XCell SureLock Mini-Cell and XCell II Blot Module (Thermo Fisher Scientific). Samples were electrophoresed under a reducing condition on NuPAGE 4–12% Bis-Tris Mini Protein Gels (Thermo Fisher Scientific). Proteins on the gel were transferred to a polyvinylidene difluoride membrane (Thermo Fisher Scientific). The membranes were blocked with 5% BSA or non-fat milk (MilliporeSigma) in TBST (Tris buffered saline with 0.1% Tween-20). After a series of washes and incubations with primary antibodies, TBST, and secondary antibodies, the immunoreactivity was visualized using SuperSignal West Dura Chemiluminescent Substrate (Thermo Fisher Scientific) and Amersham ImageQuant 800 system (Cytiva). The antibodies used were: Mouse anti-Actin antibody (C4, MAB1501, MilliporeSigma), Mouse anti-ATP5A antibody (15H4C4, abcam), Mouse anti-ND30 antibody (17D95, abcam), rabbit anti-TFAM antibody (*Liu et al., 2022*), rabbit anti-HSP60 antibody (#4870, Cell Signaling), Anti-rabbit IgG, HRP-linked Antibody (#7074, Cell Signaling), and Anti-mouse IgG, HRP-linked Antibody (#7076, Cell Signaling).

## Blue native PAGE and in-gel activity assays of ETC complexes

Mitochondria from fly larvae were isolated by homogenization and differential centrifugation following previous protocol (*Chen et al., 2015*). Solubilized protein samples from isolated mitochondria were prepared with NativePAGE Sample Prep Kit (Thermo Fisher Scientific) and the concentrations were determined by Pierce BCA protein assay (Thermo Fisher Scientific). Blue native PAGE was performed using NativePAGE 4–16% Bis-Tris gels and NativePAGE Running Buffer Kit (Thermo Fisher Scientific) according to the manufacturer's protocol. 60µg proteins for each sample were used. For in-gel activity assays, gels were incubated with one of the following solutions: Complex I buffer (5mM Tris-HCl

pH 7.4; 0.1mg/ml NADH; 2.5mg/ml Nitro Blue Tetrazolium), Complex II buffer (5mM Tris-HCl pH 7.4; 20mM sodium succinate; 0.2mM phenazine methasulfate; 2.5mg/ml Nitro Blue Tetrazolium) or Complex IV buffer (50mM sodium phosphate pH 7.2; 0.05% 3,3'-diaminobenzidine tetrahydrochloride, 50μM horse heart cytochrome *c*) at room temperature for hours, stopped by fixation with 50% methanol and 10% acetic acid for 30min and washed with 10% acetic acid. All chemicals from MilliporeSigma.

## Quantitative real-time PCR

Total genomic DNAs or RNAs were isolated using the DNeasy Blood & Tissue Kit (QIAGEN) and RNeasy Mini Kit (QIAGEN), respectively, following the manufacturer's instructions. cDNAs were synthesized by the SuperScript VILO cDNA Synthesis Kit (Thermo Fisher Scientific). Real-time PCRs were performed in triplicate using the PowerTrack SYBR Green Master Mix (Thermo Fisher Scientific), MicroAmpOptical 96-Well Reaction Plate with Barcode (Thermo Fisher Scientific), and QuantStudio 3 Real-Time PCR System (Thermo Fisher Scientific). Primers for amplifying mtDNA and nuclear DNA, as well as for measuring gene expression levels are listed in *Supplementary file 8*. The relative mtDNA levels of fly larvae or eye discs were measured in three biological replicates for each group using total DNAs extracted from 20 larvae or eye discs. The relative mRNA levels of ETC genes were measured in three biological replicates for each group using total RNAs extracted from 20 larvae.

## Prediction of protein domains, isoelectric point, net charge, and structure

Protein domains were predicted via SMART (*Schultz et al., 2000*). Protein domain isoelectric point and net charge were predicted using bioinformatic toolbox, Prot pi (https://www.protpi.ch/Calculator/ProteinTool). Protein 3D structure was predicted by AlphaFold (*Jumper et al., 2021*).

## Statistical analysis

Two-tailed Student's t-test was used for statistical analysis. The difference was considered statistically significant when p<0.05. Results are represented as mean ± SD of the number of determinations.

## Acknowledgements

We thank BDSC, Vienna Drosophila Resource Center, and Kyoto *Drosophila* Genomics and Genetics Resources for various fly lines; NHLBI Light Microscope Core and NHLBI DNA Sequencing and Genomics Core for technical assistance. This work is supported by NHLBI Intramural Research Program.

## Additional information

### Funding

| Funder | Grant reference number | Author |
| --- | --- | --- |
| National Heart, Lung, and Blood Institute | | Hong Xu |

The funders had no role in study design, data collection and interpretation, or the decision to submit the work for publication.

### Author contributions

Fan Zhang, Conceptualization, Data curation, Formal analysis, Validation, Investigation, Visualization, Methodology, Writing – original draft, Project administration, Writing – review and editing; Annie Lee, Data curation, Formal analysis, Investigation; Anna V Freitas, Data curation, Formal analysis, Investigation, Methodology; Jake T Herb, Zhe Chen, Resources; Zong-Heng Wang, Resources, Investigation, Methodology; Snigdha Gupta, Investigation; Hong Xu, Conceptualization, Resources, Data curation, Supervision, Funding acquisition, Validation, Visualization, Methodology, Writing – original draft, Project administration, Writing – review and editing

**Author ORCIDs**
Fan Zhang http://orcid.org/0000-0002-9364-6040
Hong Xu https://orcid.org/0000-0002-1423-1809

Reviewer #1 (Public review): https://doi.org/10.7554/eLife.96536.3.sa1
Reviewer #2 (Public review): https://doi.org/10.7554/eLife.96536.3.sa2
Author response https://doi.org/10.7554/eLife.96536.3.sa3

## Additional files

### Supplementary files

Supplementary file 1. List of all genes assessed in the eye screen, including gene IDs, symbols, group information, representative RNAi lines, and the Index-R of the 'RNAi-only' and 'RNAi+mitoXhoI' flies under the same heteroplasmic-mitochondrial DNAs (mtDNAs) background.

Supplementary file 2. Vertices, edges, and vertex.sort analysis information of the potential transcriptional regulatory network of nuclear-encoded mitochondrial genes.

Supplementary file 3. Gene binding profiles of 49 synergistic enhancer transcription factors (TFs), including TF-target matrix, TF binding profile summary, and list of nuclear-encoded mitochondrial genes with subgroup information as well as the counts of 49 TFs binding to each promoter.

Supplementary file 4. List of nuclear-encoded mitochondrial genes with symbols, IDs, subgroup information, and RNA-seq status.

Supplementary file 5. Differentially expressed nuclear-encoded genes in CG1603 PBac mutant flies. (a) List of differentially expressed nuclear-encoded genes in CG1603 $^{PBac}$ mutant flies compared to controls. (b) Summary of nuclear-encoded electron transport chain (ETC) genes that down-regulated more than twofold in CG1603$^{PBac}$ mutant.

Supplementary file 6. List of CG1603 peaks from modERN (model organism Encyclopedia of Regulatory Networks) ChIP-seq data, including gene annotation, group information, combined with RNA-seq analysis result of CG1603 $^{PBac}$ mutant flies.

Supplementary file 7. CG1603 binding motifs discovered by RSAT peak-motifs.

Supplementary file 8. Primer sequences for quantitative real-time PCR.

MDAR checklist

### Data availability

Sequencing data have been deposited in GEO under accession code GSE282638.All data generated or analyzed during this study are included in the manuscript and supporting files; source data files have been provided for Figures 1-8.

The following dataset was generated:

| Author(s) | Year | Dataset title | Dataset URL | Database and Identifier |
|---|---|---|---|---|
| Zhang F, Xu H | 2024 | CG1603 regulation of nuclear-encoded mitochondrial gene expression | https://www.ncbi.nlm.nih.gov/geo/query/acc.cgi?acc=GSE282638 | NCBI Gene Expression Omnibus, GSE282638 |

The following previously published datasets were used:

| Author(s) | Year | Dataset title | Dataset URL | Database and Identifier |
|---|---|---|---|---|
| modENCODEProject | 2016 | TF ChIP-seq_ab | https://www.encodeproject.org/experiments/ENCSR140RMO | ENCODE, ENCSR140RMO |

*Continued on next page*

*Continued*

| Author(s) | Year | Dataset title | Dataset URL | Database and Identifier |
|---|---|---|---|---|
| modENCODEProject | 2019 | TF ChIP-seq_abd-A | https://www.encodeproject.org/experiments/ENCSR609JDR | ENCODE, ENCSR609JDR |
| modENCODEProject | 2016 | TF ChIP-seq_achi | https://www.encodeproject.org/experiments/ENCSR959SWC | ENCODE, ENCSR959SWC |
| modENCODEProject | 2016 | TF ChIP-seq_bdp1 | https://www.encodeproject.org/experiments/ENCSR504AYW | ENCODE, ENCSR504AYW |
| modENCODEProject | 2018 | TF ChIP-seq_bigmax | https://www.encodeproject.org/experiments/ENCSR224YOT | ENCODE, ENCSR224YOT |
| modENCODEProject | 2016 | TF ChIP-seq_CG10631 | https://www.encodeproject.org/experiments/ENCSR751FSX | ENCODE, ENCSR751FSX |
| modENCODEProject | 2016 | TF ChIP-seq_CG11398 | https://www.encodeproject.org/experiments/ENCSR486MQI | ENCODE, ENCSR486MQI |
| modENCODEProject | 2018 | TF ChIP-seq_CG12299 | https://www.encodeproject.org/experiments/ENCSR028DHQ | ENCODE, ENCSR028DHQ |
| modENCODEProject | 2018 | TF ChIP-seq_CG12391 | https://www.encodeproject.org/experiments/ENCSR483VVZ | ENCODE, ENCSR483VVZ |
| modENCODEProject | 2018 | TF ChIP-seq_CG15011 | https://www.encodeproject.org/experiments/ENCSR608VFH | ENCODE, ENCSR608VFH |
| modENCODEProject | 2019 | TF ChIP-seq_CG1603 | https://www.encodeproject.org/experiments/ENCSR680NFF | ENCODE, ENCSR680NFF |
| modENCODEProject | 2016 | TF ChIP-seq_CG16863 | https://www.encodeproject.org/experiments/ENCSR848GYF | ENCODE, ENCSR848GYF |
| modENCODEProject | 2018 | TF ChIP-seq_CG17806 | https://www.encodeproject.org/experiments/ENCSR118TAH | ENCODE, ENCSR118TAH |
| modENCODEProject | 2019 | TF ChIP-seq_CG17829 | https://www.encodeproject.org/experiments/ENCSR810MLF | ENCODE, ENCSR810MLF |
| modENCODE | 2018 | TF ChIP-seq_CG2116 | https://www.encodeproject.org/experiments/ENCSR456UZW | ENCODE, ENCSR456UZW |

*Continued on next page*

*Continued*

| Author(s) | Year | Dataset title | Dataset URL | Database and Identifier |
|---|---|---|---|---|
| modENCODEProject | 2018 | TF ChIP-seq_CG4707 | https://www.encodeproject.org/experiments/ENCSR302AQS | ENCODE, ENCSR302AQS |
| modENCODEProject | 2016 | TF ChIP-seq_CG4854 | https://www.encodeproject.org/experiments/ENCSR860FUA | ENCODE, ENCSR860FUA |
| modENCODEProject | 2017 | TF ChIP-seq_CG6254 | https://www.encodeproject.org/experiments/ENCSR233UXG | ENCODE, ENCSR233UXG |
| modENCODEProject | 2018 | TF ChIP-seq_CG7987 | https://www.encodeproject.org/experiments/ENCSR681RZY | ENCODE, ENCSR681RZY |
| modENCODEProject | 2018 | TF ChIP-seq_CG9437 | https://www.encodeproject.org/experiments/ENCSR965XSS | ENCODE, ENCSR965XSS |
| modENCODEProject | 2017 | TF ChIP-seq_CG9727 | https://www.encodeproject.org/experiments/ENCSR624PKN | ENCODE, ENCSR624PKN |
| modENCODEProject | 2016 | TF ChIP-seq_CHES-1-like | https://www.encodeproject.org/experiments/ENCSR791TWT | ENCODE, ENCSR791TWT |
| modENCODEProject | 2016 | TF ChIP-seq_Crg-1 | https://www.encodeproject.org/experiments/ENCSR596YNC | ENCODE, ENCSR596YNC |
| modENCODEProject | 2017 | TF ChIP-seq_CTCF | https://www.encodeproject.org/experiments/ENCSR661BEZ | ENCODE, ENCSR661BEZ |
| modENCODEProject | 2016 | TF ChIP-seq_E(bx) | https://www.encodeproject.org/experiments/ENCSR559AJG | ENCODE, ENCSR559AJG |
| modENCODEProject | 2019 | TF ChIP-seq_Ets96B | https://www.encodeproject.org/experiments/ENCSR572JUN | ENCODE, ENCSR572JUN |
| modENCODEProject | 2017 | TF ChIP-seq_ewg | https://www.encodeproject.org/experiments/ENCSR444JKK | ENCODE, ENCSR444JKK |
| modENCODEProject | 2018 | TF ChIP-seq_Fer3 | https://www.encodeproject.org/experiments/ENCSR367SBM | ENCODE, ENCSR367SBM |
| modENCODEProject | 2016 | TF ChIP-seq_ftz-f1 | https://www.encodeproject.org/experiments/ENCSR741KZZ | ENCODE, ENCSR741KZZ |

*Continued*

| Author(s) | Year | Dataset title | Dataset URL | Database and Identifier |
|---|---|---|---|---|
| modENCODEProject | 2016 | TF ChIP-seq_ind | https://www.encodeproject.org/experiments/ENCSR486EPD | ENCODE, ENCSR486EPD |
| modENCODEProject | 2016 | TF ChIP-seq_Jra | https://www.encodeproject.org/experiments/ENCSR471GSA | ENCODE, ENCSR471GSA |
| modENCODEProject | 2017 | TF ChIP-seq_Lhr | https://www.encodeproject.org/experiments/ENCSR272DEA | ENCODE, ENCSR272DEA |
| modENCODEProject | 2016 | TF ChIP-seq_luna | https://www.encodeproject.org/experiments/ENCSR770AUN | ENCODE, ENCSR770AUN |
| modENCODEProject | 2016 | TF ChIP-seq_Met | https://www.encodeproject.org/experiments/ENCSR363SHZ | ENCODE, ENCSR363SHZ |
| modENCODEProject | 2016 | TF ChIP-seq_Myb | https://www.encodeproject.org/experiments/ENCSR393AKW | ENCODE, ENCSR393AKW |
| modENCODEProject | 2017 | TF ChIP-seq_Myc | https://www.encodeproject.org/experiments/ENCSR191VCQ | ENCODE, ENCSR191VCQ |
| modENCODEProject | 2019 | TF ChIP-seq_nerfin-1 | https://www.encodeproject.org/experiments/ENCSR335NNR | ENCODE, ENCSR335NNR |
| modENCODEProject | 2018 | TF ChIP-seq_Nnk | https://www.encodeproject.org/experiments/ENCSR165KWP | ENCODE, ENCSR165KWP |
| modENCODEProject | 2016 | TF ChIP-seq_p53 | https://www.encodeproject.org/experiments/ENCSR808XNJ | ENCODE, ENCSR808XNJ |
| modENCODEProject | 2017 | TF ChIP-seq_pan | https://www.encodeproject.org/experiments/ENCSR033IIP | ENCODE, ENCSR033IIP |
| modENCODEProject | 2019 | TF ChIP-seq_pros | https://www.encodeproject.org/experiments/ENCSR682YQM | ENCODE, ENCSR682YQM |
| modENCODEProject | 2017 | TF ChIP-seq_REPTOR-BP | https://www.encodeproject.org/experiments/ENCSR271ZJI | ENCODE, ENCSR271ZJI |
| modENCODEProject | 2016 | TF ChIP-seq_salr | https://www.encodeproject.org/experiments/ENCSR042XCV | ENCODE, ENCSR042XCV |

*Continued on next page*

*Continued*

| Author(s) | Year | Dataset title | Dataset URL | Database and Identifier |
|---|---|---|---|---|
| modENCODEProject | 2019 | TF ChIP-seq_Stat92E | https://www.encodeproject.org/experiments/ENCSR290OJD | ENCODE, ENCSR290OJD |
| modENCODEProject | 2017 | TF ChIP-seq_su(Hw) | https://www.encodeproject.org/experiments/ENCSR761TCG | ENCODE, ENCSR761TCG |
| modENCODEProject | 2019 | TF ChIP-seq_trem | https://www.encodeproject.org/experiments/ENCSR104XOO | ENCODE, ENCSR104XOO |
| modENCODEProject | 2016 | TF ChIP-seq_Trl | https://www.encodeproject.org/experiments/ENCSR629WQT | ENCODE, ENCSR629WQT |
| modENCODEProject | 2016 | TF ChIP-seq_Xbp1 | https://www.encodeproject.org/experiments/ENCSR698IEF | ENCODE, ENCSR698IEF |
| modENCODEProject | 2017 | TF ChIP-seq_YL-1 | https://www.encodeproject.org/experiments/ENCSR396LMJ | ENCODE, ENCSR396LMJ |

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
