## [Editor Report · eLife Assessment]

This study's findings substantially advance our understanding of an **important** aspect of mitochondrial metabolism. The data are **compelling** and the study is well executed. The work is relevant to all who are interested in the biogenesis of mitochondria.

---

## [Referee Report · Reviewer #1 (Public review)]

In this manuscript, Zhang et al. report a genetic screen to identify novel transcriptional regulators that coordinate mitochondrial biogenesis. They performed an RNAi-based modifier screen wherein they systematically knocked down all known transcription factors in the developing *Drosophila* eye, which was sensitized and had decreased mitochondrial DNA content. Through this screen, they identify CG1603 as a potential regulator of mitochondrial volume. They show that protein levels of mitochondrial proteins like TFAM, SDHA, and other mitochondrial proteins and mtDNA content are downregulated in CG1603 mutants. RNA-Seq and ChIP-Seq further show that CG1603 binds to the promoter regions of several known nuclear-encoded mitochondrial genes and regulates their expression. Finally, they also identified YL-1 as an upstream regulator of CG1603. Most studies have focused on PGC-1α as a master regulator of mitochondrial biogenesis. which seems to be a context-dependent regulator. Also, PGC-1α mediated regulation does not explain the regulation of 1100 genes that are required for mitochondrial biogenesis. Therefore, identifying new regulators in this work is crucial for the advancement of our understanding of mitochondrial biogenesis.

---

## [Referee Report · Reviewer #2 (Public review)]

Summary:

In this study, the authors identified nuclear genome-encoded transcription factors that regulate mtDNA maintenance and mitochondrial biogenesis. They started with an RNAi screening in developing *Drosophila* eyes with reduced mtDNA content and identified several putative candidate genes. Subsequently, using ChIP-seq data, they built a potential regulatory network that seems to govern mitochondrial biogenesis. Next, they focused on a candidate gene, CG1603 /clifford, for further characterization. Based on the expression of different markers, such as TFAM and SDHA, in RNAi and overexpression clones in the midgut, they argued that CG1603 promotes mitochondrial biogenesis and the expression of ETC complex genes. They used a CG1603 mutant to show reduced mtDNA and mitochondrial protein levels. Clonal analyses showed a reduction in mitochondrial biogenesis and membrane potential upon loss of CG1603. They further showed that the protein is localized to the mitochondria, and binds to polytene chromosomes in the salivary gland. Based on the RNA-seq results from the mutants and the ChIP data, the authors argued that the nucleus-encoded mitochondrial genes are downregulated >2 folds in the CG1603 mutants and that the regulatory elements bound by CG1603 are related to ETC biogenesis. Finally, they showed that YL-1, another candidate in the network, is an upstream regulator of CG1603. The screening strategy was well-designed, and the follow-up experiments were nicely executed.

Comments on revisions:

The authors have addressed my previous comments satisfactorily.

---

## [Author Response]

The following is the authors’ response to the original reviews.

**Public Reviews:**

**Reviewer #1 (Public Review):**
Summary:In this manuscript, Zhang et al. report a genetic screen to identify novel transcriptional regulators that could coordinate mitochondrial biogenesis. They performed an RNAi-based modifier screen wherein they systematically knocked down all known transcription factors in the developing *Drosophila* eye, which was already sensitised and had decreased mitochondrial DNA content. Through this screen, they identify CG1603 as a potential regulator of mitochondrial content. They show that protein levels of mitochondrial proteins like TFAM, SDHA, and other mitochondrial proteins and mtDNA content are downregulated in CG1603 mutants. RNA-Seq and ChIP-Seq further show that CG1603 binds to the promoter regions of several known nuclear-encoded mitochondrial genes and regulates their expression. Finally, they also identified YL-1 as an upstream regulator of CG1603. Overall, it is a very important study as our understanding of the regulation of mitochondrial biogenesis remains limited across metazoans. Most studies have focused on PGC-1α as a master regulator of mitochondrial biogeneis, which seems a context-dependent regulator. Also, PGC-1α mediated regulation could not explain the regulation of 1100 genes that are required for mitochondrial biogenesis. Therefore, identifying a new regulator is crucial for understanding the overall regulation of mitochondrial biogenesis.
**Reviewer #2 (Public Review):**
Summary:In this study, the authors aim to identify the nuclear genome-encoded transcription factors that regulate mtDNA maintenance and mitochondrial biogenesis. They started with an RNAi screening in developing *Drosophila* eyes with reduced mtDNA content and identified a number of putative candidate genes. Subsequently, using ChIP-seq data, they built a potential regulatory network that could govern mitochondrial biogenesis. Next, they focused on a candidate gene, CG1603, for further characterization. Based on the expression of different markers, such as TFAM and SDHA, in the RNAi and OE clones in the midgut cells, they argue that CG1603 promotes mitochondrial biogenesis and the expression of ETC complex genes. Then, they used a mutant of CG1603 and showed that both mtDNA levels and mitochondrial protein levels were reduced. Using clonal analyses, they further show a reduction in mitochondrial biogenesis and membrane potential upon loss of CG1603. They made a reporter line of CG1603, showed that the protein is localized to the mitochondria, and binds to polytene chromosomes in the salivary gland. Based on the RNA-seq results from the mutants and the ChIP data, the authors argue that the nucleus-encoded mitochondrial genes that are downregulated >2 folds in the CG1603 mutants and that are bound by CG1603 are related to ETC biogenesis. Finally, they show that YL-1, another candidate in the network, is an upstream regulator of CG1603.Strengths:This is a valuable study, which identifies a potential regulator and a network of nucleus-encoded transcription factors that regulate mitochondrial biogenesis. Through in-vivo and in-vitro experimental evidence, the authors identify the role of CG1603 in this process. The screening strategy was smart, and the follow-up experiments were nicely executed.Weaknesses:Some additional experiments showing the effects of CG1603 loss on ETC integrity and functionality would strengthen the work.
**Recommendations for the authors:**

**Reviewer #1 (Recommendations For The Authors):**
(1) Fig 3F: SDHA levels are severely downregulated in CG1603 RNAi clones. Therefore, estimating mitochondrial volume based on the SDHA reporter might be misleading. I suggest the authors perform this experiment with an independent marker of mitochondria, like mitoTracker Green or other dyes. I also suggest checking for mitochondrial number/quantity/size by electron microscopy.

Even though being downregulated, the SDHA-mNeon signal in EC clones clearly outlined mitochondria and the overall mitochondrial network, allowing us to quantify the total mitochondrial volume. Examining mitochondrial number/quantity/size by electron microscopy would further strengthen this statement, and we will consider it in future studies.

(2) The authors might comment on whether there was any decrease in the volume of CG1603i clone cells. And whether this was taken into account while normalising the mitochondrial volume.

The size/volume of CG1603i clone cells were indeed decreased, which was considered while normalizing the mitochondrial volume. We clarified this point in methods section (page 18, line 511-512 (revised version page 18, line 515-517)).

(3) Line 230-234: Collectively, these results demonstrate that CG1603 promotes the expression of both nuclear and mtDNA-encoded ETC genes and boosts mitochondrial biogenesis. CG1603 RNAi produced very few EC clones, consistent with the notion that mitochondrial respiration is necessary for ISCs differentiation.(4) Quantifying the number of EC clone cells observed might help support this statement.

This is a great point. We quantified the number of EC clone cells, and the data was included in the revised Figure 3—figure supplement.

(5) Figure 5: The intensity of MTGreen in CH1603 clones seems comparable to that in control cells, at least visually. Since the authors claim a reduction in mitochondrial volume in CG1603 mutants, it is crucial to estimate mitochondrial volume based on MTGreen intensity in mutant and control cells.

There are two types of clones shown in Figure 5: germ cell clones including all 16 germ cells in the same egg chamber and follicle cell clones. We highlight these two types of clones in the revised Figure 5, to emphasize this point. The total MT Green intensity in both germ cell and follicle cell *CG1603PBac* clones were reduced, compared to germ cells in adjacent egg chambers and adjacent follicle cells in the same egg chamber, respectively. We included the quantification of MTGreen intensity in the revised Figure 5—figure supplement C. Examining mitochondrial number/quantity/size by electron microscopy would further strengthen this statement, and we will consider it in future studies.

(6) Figure 8: It would be interesting to know what happens to steady-state mtDNA levels during YL-1 knockdown. If decreased, could overexpressing CG1603 in YL-1 knockdown cells rescue the phenotype?

YL-1 knockdown reduced steady-state mtDNA levels in eyes, and overexpressing CG1603 restored mtDNA level in YL-1 knockdown cells. These results are included in the revised *Figure 8-figure supplement C*.

Minor comments:(7) The paper is lucidly written, but there are minor typos in several places. The authors might proofread it to remove these errors.

We corrected typos and other minor errors in the manuscript.

(8) Quantification for Figure 8 - Supplementary needs to be included.

We performed the quantification, and the result is shown in Figure 8—figure supplement B.

**Reviewer #2 (Recommendations For The Authors):**
(1) In lines 275-276 and Figure 6E, the authors mention that more than 800 nuclear-encoded mitochondrial genes were reduced by >2-folds in CG1603 mutants. One gene related to mitochondrial replication and three genes related to mtDNA transcription were among them. Was TFAM one of these candidates? What were the reduction levels of TFAM mRNA in RNA seq results? Can the author confirm it via RT-PCR?

In RNAseq analyses, TFAM was differentially expressed with a log2 Fold-Change of “ -0.74”, corresponding to ~1.6-fold decrease, and hence was not one of these candidates that were down-regulated more than two folds in CG1603 mutant. Per reviewer’s suggestion, we carried out RT-PCR and found TFAM was downregulated about 2-fold in CG1603 mutant. We included this result in the revised Figure 6F and listed all differentially expressed genes in Supplementary file 5a.

(2) In many places, the authors argued about the role of CG1603 in ETC biogenesis. Also, the RNA-seq data shows that 64 genes related to the ETC complex were reduced by > 2-fold in CG1603 mutant. Therefore, it would be critical to expand a little on this aspect. For example, what are these genes and related to which of the ETC complex? Can the authors show the reduced levels of some of the candidate genes from each complex via RT-PCR?

We listed all ETC genes that were down-regulated more than two folds in CG1603 mutant in a separate sheet in Supplementary file 5b. We further validated the reduced expression of ETC genes by RT-PCR on three randomly selected candidate genes from each complex. The result is included in the revised Figure 6F.

(3) To make their argument solid on the role of CG1603 on ETC biogenesis, it is important to show the assembly/integrity of ETC complexes as well as the functionality/activity of the ETC complexes in CG1603 mutants.

We purified mitochondria, and assayed assembly/integrity of three ETC complexes (Complex I, II and IV) and their activities, using blue native PAGE analysis and in gel activity analysis, respectively. The amount of these three complexes, and accordingly, their activities were all markedly reduced in CG1603 mutant compared to wt. The result is included as Figure 4—figure supplement A.

(4) CG1603 has already been named as cliff. Why do the authors not use this name, or alternatively propose one?

We thank the reviewer for the note. The CG1603 has not been named as cliff when we were preparing this manuscript.

(5) In lines 230-231, based on the TFAM-GFP and SDHA-mNG levels, the authors claim that "these results demonstrate that CG1603 promotes the expression of both nuclear and mtDNA-encoded ETC genes..." The authors may tone down this statement since it sounds overstating. It would be prudent to claim that a subset of genes are regulated by CG1603.

We appreciate the reviewer’s suggestion. We revised the text to tone down this statement (page 8, line 201; page 9, line 229-230).